# Modelling of negative equivalent magnetic reluctance structure and its application in weak-coupling wireless power transmission

Yuanxi Chen [1], Shuangxia Niu [1] ✉, Weinong Fu [2] & Hongjian Lin[3]

In weak-coupling wireless power transmission, increasing operating frequency, and incorporating metamaterials, resonance structures or ferrite cores have been explored as effective solutions to enhance power efficiency. However, these solutions present significant challenges that need to be addressed. The increased operating frequency boosts ferrite core losses when it exceeds the working frequency range of the material. Existing metamaterial-based solutions present challenges in terms of requiring additional space for slab installation, resulting in increased overall size. In addition, limitations are faced in using Snell's law for explaining the effects of metamaterial-based solutions outside the transmission path, where the magnetic field can not be reflected or refracted. To address these issues, in this work, the concept of a negative equivalent magnetic reluctance structure is proposed and the metamaterial theory is extended with the proposed magnetic reluctance modelling method. Especially, the negative equivalent magnetic reluctance structure is effectively employed in the weak-coupling wireless power transfer system. The proposed negative equivalent magnetic reluctance structure is verified by the stacked negative equivalent magnetic reluctance structure-based transformer experiments and two-coil mutual inductance experiments. Besides, the transmission gain, power experiments and loss analysis experiments verify the effectiveness of the proposed structure in the weak-coupling wireless power transfer system.

Wireless power transfer (WPT) technology[1–3] is a fast-growing charging solution for electric vehicles[4], sensors[5,6], home automation[7], and medical and biological applications[8–10]. The operating frequency of the WPT systems typically ranges from kHz to MHz, largely dependent on the coupling coefficient of the coils. The coupling coefficient of the generalized 85 kHz WPT system[11,12] is usually larger than 0.15, to ensure a qualified transfer efficiency of the system. While for the weak-coupling WPT system[13–16], the coupling coefficient is much lower than the generalized solution, which cannot operates with high efficiency in the kHz frequency region. Generalized solutions employ the magnetic core or increased operation frequency to enhance efficiency. The

magnetic ferrite core with high permeability can reduce the total magnetic reluctance, thereby increasing the mutual inductance and coupling between the coils[17]. However, the hysteresis loss of the iron-oxide ferrite will boost when the system operating frequency exceeds the working frequency range of ferrite materials, leading to a decrease in the efficiency of the weak-coupling WPT system[18]. Hence, a conventional weak-coupling WPT system cannot effectively incorporate both a generalized ferrite core and operating at high frequencies.

To address this issue, researchers have been working on developing specialized core materials and designs for these high-frequency, weak-coupling WPT systems. The designed $Nd_x Fe_{1-x} N_y$ material[19] as

[1]Department of Electrical and Electronic Engineering, The Hong Kong Polytechnic University, 999077 Hong Kong, China. [2]Faculty of Computer Science and Control Engineering, Shenzhen University of Advanced Technology, Shenzhen 518107, China. [3]Department of Electrical Engineering, City University of Hong Kong, 999077 Hong Kong, China. ✉e-mail: eesxniu@polyu.edu.hk

the magnetic core in a 13.56 MHz system increases the inductance from 0.69 to 1.15 µH. The cap-shaped back yoke topology[20] for the MHz WPT system explores the impact of different core materials, such as Ni-Zn, Fe-Si, and amorphous, on efficiency enhancement. The results show an efficiency improvement ranging from 0.7 to 1.2%.

Employing resonance coil[21-23] is another widely used solution for efficiency enhancement in weak-coupling WPT systems. A dual-intermediate resonant coil[21] design achieved an efficiency of 72.4% at 4.63 MHz. A 13.56 MHz WPT system with multiple coupling paths[22] also demonstrates increased efficiency. The superconductivity resonance coils[23] has been shown to increase the efficiency of the system from 17.5 to 49.7%. Apart from the above-mentioned solutions, metamaterials and metasurfaces have also been investigated to enhance the efficiency of weak-coupling WPT systems[24-31] as well as improve the misalignment tolerance[32,33]. The key is to design and achieve either a negative permeability to refract the electromagnetic field[24-29] or near-zero permeability to reflect the electromagnetic field[30,31], thereby increasing the flux on the receiver coil and enhancing the overall efficiency.

However, for employing unconventional core materials, the effectiveness of efficiency enhancement is limited[19,20]. Additionally, due to the positive permeability of ferrite materials, the corresponding magnetic reluctance always remains positive, regardless of optimization and design. Consequently, in terms of magnetic reluctance reduction for weak-coupling WPT systems, the ferrite materials are inherently weaker than the metamaterials with negative permeability in efficiency enhancement. Given the reasons above, employing metamaterials is considered a potentially ideal solution for a weak-coupling WPT system. Nevertheless, the application of existing metamaterial-based solutions is not only limited by the low practicability but also the theoretical issue. Firstly, the metamaterials[24-31] occupy additional space beyond the coils, which significantly increases the overall size of the weak-coupling WPT systems. A transmitter-embedded metasurface[34] can solve the space-occupying issue. Secondly, the existing theory based on Snell's law cannot properly explain the effect of metamaterials outside the transmission path, which cannot reflect or refract the magnetic field generated by the transmitter coil, i.e. the metamaterial is installed in the receiver coil. Besides, the generalized metamaterial/metasurface requires a quantity of units to generate a homogeneous material. This design limits the quality factor of the units, as well as increases the corresponding loss, making the resonator can only effectively operate at relatively high frequency with large size.

To address the aforementioned issues, the concept of a negative equivalent magnetic reluctance (NEMR) structure and its modelling method, as well as its application in a weak coupling WPT system are proposed and verified. This design installs the NEMR structure in both the transmitter and receiver coils, aiming to increase the mutual inductance and enhance efficiency by reducing the total magnetic reluctance based on negative permeability and magnetic reluctance. The key of this design is to indicate a negative equivalent magnetic reluctance with a design of L/C combination at a specific frequency range and regulate the magnetomotive force (MMF), making the MMF generated by the NEMR structure close to that generated by the transmitter. If the vector summation of MMFs is larger than that of the transmitter, the flux of the magnetic circuit will increase. Correspondingly, the structure indicates a 'negative equivalent magnetic reluctance'. A general comparison is given in Table S1 (see Supplementary Note 1 for a detailed comparison), including a summary of the technology and key specifications.

The main contributions of this paper are summarized as follows. (i) A magnetic reluctance-based modelling method is proposed to enhance the metamaterial theoretical analysis, extending the negative permeability effect to the negative magnetic reluctance effect on the multi-coil system. (ii) The impact of magnetic reluctance, mutual inductance, and permeability on the efficiency of the weak-coupling WPT system is investigated. (iii) The concept of negative equivalent magnetic reluctance is proposed and verified, and the relationship between magnetic reluctance and frequency is studied. (iv) This proposed design enhances the efficiency of the weak-coupling WPT system without sacrificing space occupation while increasing the versatility compared to conventional metamaterial/metasurface-based solutions.

The schematic configuration of the proposed NEMR structure based weak-coupling WPT system is given in Fig. 1. The system consists of a receiver coil, transmitter coil, and two coil-embedded NEMR structures. The installation position of the proposed NEMR structure takes into account the practicality of the system, as it does not require any additional space apart from the area occupied by the transmitter and receiver coils. (The parameters and configuration of the NEMR structure-based WPT system are given in Fig. S1 and Table S2 of Supplementary Note 2)

## Result
### Electromagnetic analysis
The magnetic field strength **H** is directly proportional to the current **I** in a conductor (The detailed analysis is given in Supplementary Note 3). By observing the magnetic field strength **H** of the system with constant load and input power, the effect of the NEMR structure on the weak-coupling WPT systems can be evaluated. The magnetic field distributions before and after introducing the NEMR structure are shown in Fig. 2a–d, Considering the load of those four systems and the input power is constant, the higher current in the receiver coil indicates a higher receiver power and higher efficiency. As shown in Fig. 2, the WPT system with the dual coil-embedded NEMR structure has the best performance (highest magnetic field strength **H** around the receiver), followed by the system with transmitter-embedded, receiver-embedded NEMR structure, as well as that without the NEMR structure. The NEMR structures increase the magnetic field intensity **H** around the receiver coil to a different extent, which is directly connected to the power transfer efficiency of the weak-coupling WPT system. Besides, excepting the magnetic field strength **H** on the transmission path, the magnetic field strength **H** around the receiver coil on the plan view is shown in Fig. 2e–h to further study the impact of the NEMR structure on the weak-coupling WPT system. The results indicate that the weak-coupling WPT system with the dual coil-embedded NEMR structure has the best performance, which has the same trend reported in Fig. 2a–d.

### Negative magnetic reluctance property verification
To evaluate the equivalent magnetic reluctance of the NEMR structure that can be achieved under a specific frequency region, experiments of the transformer with a stacked magnetic core, consisting of different layers of the NEMR structure, are conducted. The number of turns of the primary side and secondary side of the transformer is selected as 15. The system configuration, equivalent electric circuit, and experiment platform of the designed transformer are shown in Fig. 3a–c.

In the no-load operating condition, the secondary current $i_s$ is considered as zero. The relationship between input voltage $U_s$ and output voltage $U_o$ can be found as follows. (The detailed analysis see Supplementary Note 4).

$$\frac{U_o}{U_s} \approx k \frac{j\omega N^2}{j\omega N^2 + R_p R_{mt}} \tag{1}$$

where $k$ is the coupling coefficient of the primary coil and secondary coil. $R_{mt}$ is the total magnetic reluctance of the NEMR structure-based transformer. $R_p$ is the resistance of the primary coil.

As the primary resistance $R_p$ is greater than zero and the coupling coefficient $k$ is less than 1, only if the magnetic reluctance $R_{mt}$ is

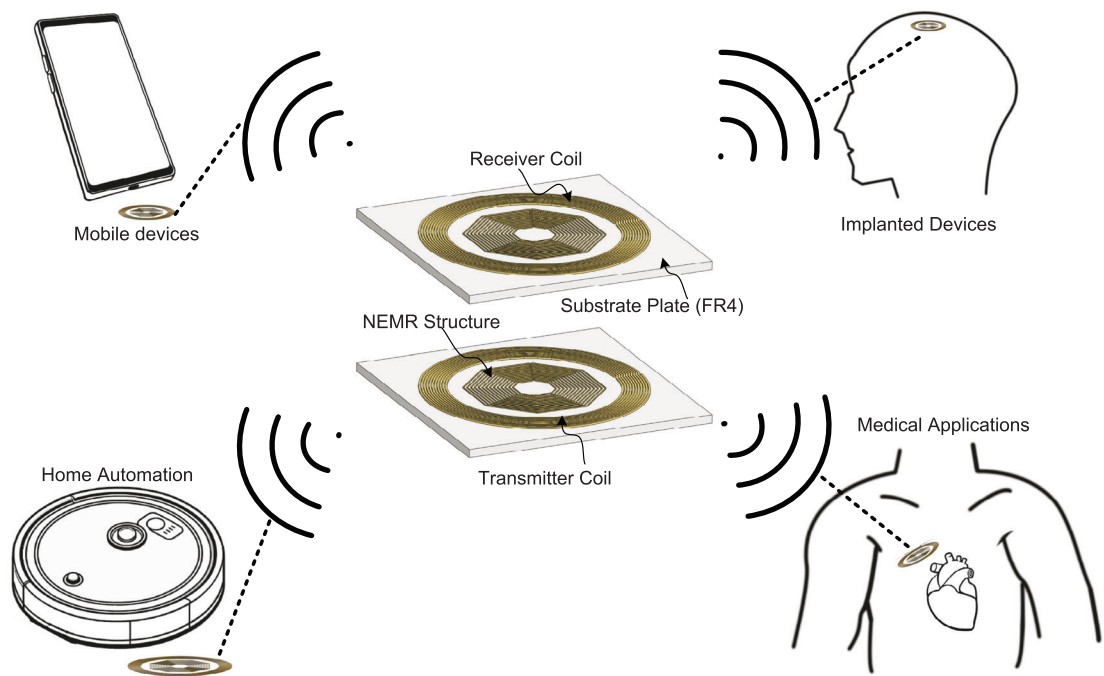

**Fig. 1 | The configuration and applications of the NEMR structure-based WPT system.** The system consits of a transmitter coil, receiver coil, and two coil-embedded NEMR structures. The typical applications of this NEMR structure-based WPT system include the mobile devices, implanted devices, home automation and medicatl applications.

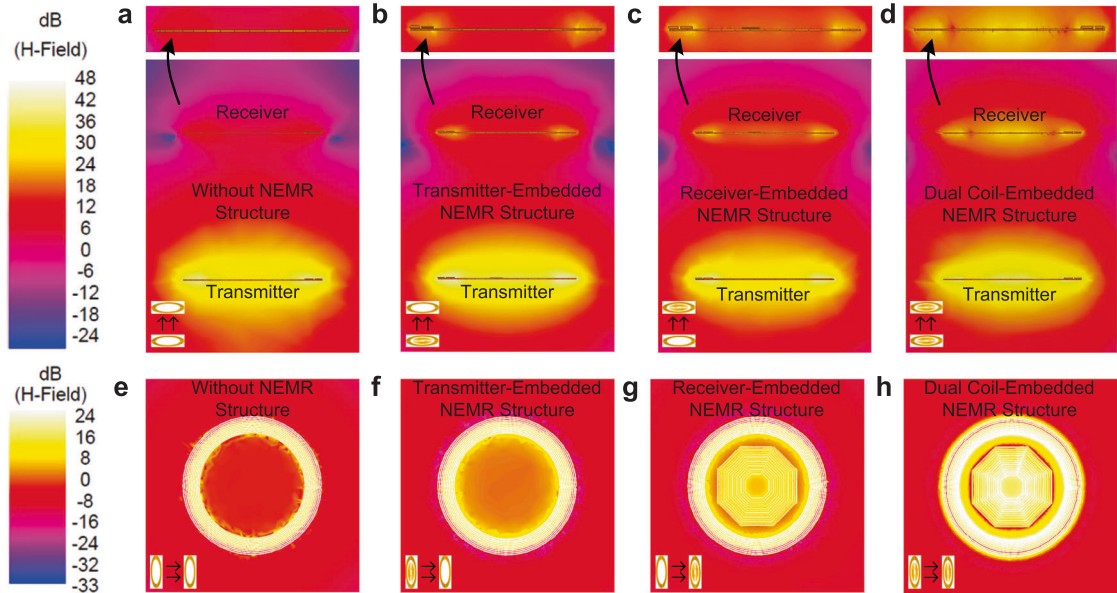

**Fig. 2 | The electromagnetic distribution of the weak-coupling WPT systems.** **a** Transmission path without NEMR structure. **b** Transmission path with the transmitter-embedded NEMR structure. **c** Transmission path with the receiver-embedded NEMR structure. **d** Transmission path with dual coil-embedded NEMR structure. **e** Receiver without NEMR structure. **f** Receiver with the transmitter-embedded NEMR structure. **g** Receiver with the receiver-embedded NEMR structure. **h** Receiver with dual coil-embedded NEMR structure.

negative, the proportion between the output voltage $U_o$ and primary voltage $U_s$ could be larger than 1. (The detailed analysis and proof see Supplementary Note 4).

The configuration, equivalent electric circuit, verification system, and prototype of the transformer with the stacked core of NEMR structure are presented in Fig. 3a–d. The voltage proportions of the transformer with the NEMR structure core or air core under different frequencies are conducted and the experimental results are shown in Fig. 3e–j. With the core of air, the voltage proportion of the secondary

coil and primary coil is 0.613, see Fig. 3h. Conversely, under the frequency of 6.7 MHz, the voltage proportion of the NEMR structure will boost to 8.31 V, which means that the NEMR structure indicates negative equivalent magnetic reluctance. The voltage proportion of the transformer with the proposed NEMR structure is always larger than that without the NEMR structure, which peaks at 6.7 MHz approximately. Besides, to verify that the magnetic reluctance of the stacked NEMR structure is impacted by the volume, the experiments of the transformer with different layers (associated with the volume) of

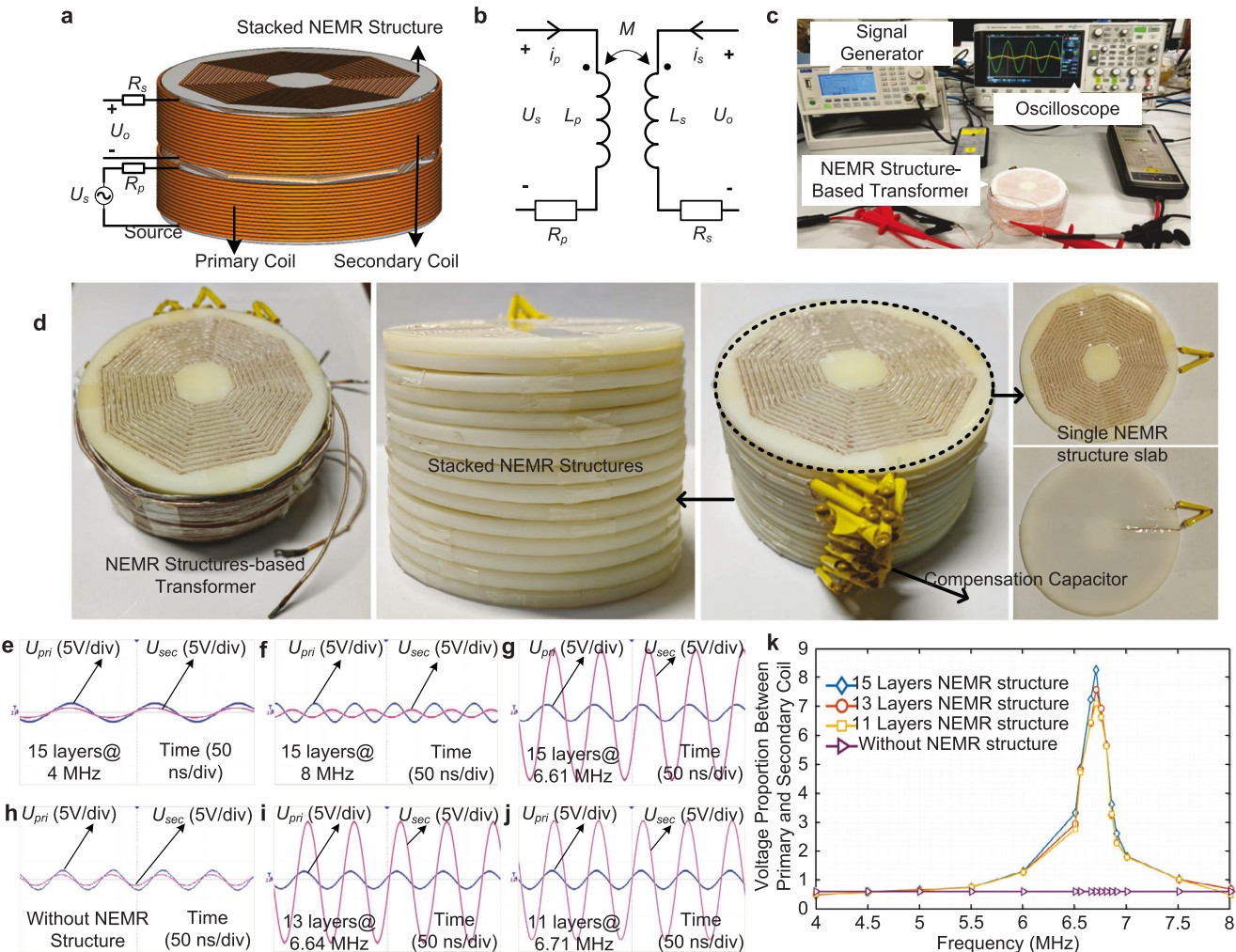

**Fig. 3 | The transformer-based system for negative magnetic reluctance verification and results. a** NEMR structure-based transformer. **b** Equivalent circuit of the transformer. **c** Verification system. **d** The schematic diagram of the stacked NEMR structures in transformer. **e** Voltage proportion with 15-layer NEMR structure at 4 MHz. **f** Voltage proportion with 15-layer NEMR structure at 8 MHz. **g** Voltage proportion with 15-layer NEMR structure at 6.61 MHz. **h** Voltage proportion without NEMR structure. **i** Voltage proportion with 13-layer NEMR structure at 6.64 MHz. **j** Voltage proportion with 11-layer NEMR structure at 6.71 MHz. **k** The voltage proportion versus the operating frequency with or without the NEMR structure.

the stacked NEMR structure core are conducted. Results in Fig. 3i, j indicate that the voltage proportion increases with the increase in the number of layers (volume) of the stacked NEMR structure. Compared to the stacked NEMR structure with 15 layers, the peak voltage proportion of the transformer with 13 and 11 layers will decrease from 8.31 to 7.62 and 7.15, respectively.

**Mutual inductance improvement verification**

The efficiency of the weak-coupling WPT system is greatly impacted by the mutual inductance $M$ The mutual inductance $M$ can be enhanced with negative magnetic reluctance. (Detailed analysis and proof see Supplementary Note 5). In the weak-coupling WPT system, if the receiver coil is open-circuit and both receiver/transmitter coils are without compensation capacitors, the receiver current $I_p$ is zero. The induced voltage $U_{sec}$ in the receiver coil is determined by the mutual inductance $M$ and the angular frequency $\omega$ of the weak-coupling WPT system, expressed as,

$$j\omega M I_p = U_{sec} \tag{2}$$

The NEMR can effectively increase the secondary voltage with a lower primary current under different transfer distances, see Fig. 4a–f. The coupling coefficient versus transfer distance of the WPT system

with and without the proposed NEMR structure is computed and given in Fig. 4g. As shown in Fig. 4g, the system with the dual NMER structure has the highest coupling coefficient among those systems. Based on the above-mentioned results, the proposed NEMR structure can effectively increase the mutual inductance $M$ between the coils.

**Transmission gain verification**

The verification platform consists of a spectrum analyzer and vector network analyzer (VNA) SVA-1032X, which is used to verify the transmission gain and is shown in Fig. 5a. The compensation topology is selected as the series-series topology.

As described in the generalized solutions, the entire weak-coupling WPT system with and without the NEMR structure is regarded as a two-port network[24,25]. The transmission gain of the system is obtained by the reflection coefficient S$_{11}$ and the forward transmission coefficient S$_{21}$. In generalized WPT verification systems, the transmitter and receiver coil are connected to ports 1, and 2 of the VNA. The coil-to-coil efficiency $\eta_{coil}$ under ideal condition[30,31] is defined as,

$$\eta_{coil} = S_{21}^2 \tag{3}$$

where S$_{21}$ is the transmission coefficient. (The analysis and formula derivation see Supplementary Note 6).

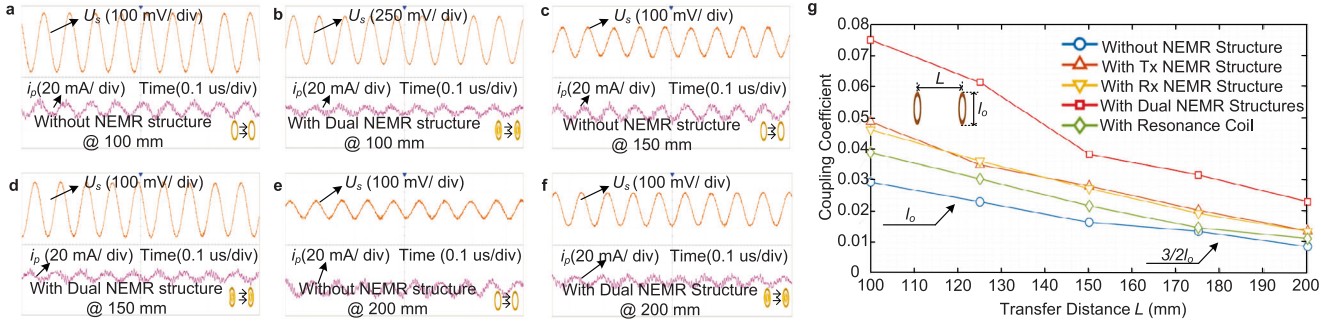

**Fig. 4 | The mutual-inductance experiments and results. a** Primary current versus secondary voltage of WPT system without NEMR structure at 100 mm. **b** Primary current versus secondary voltage of WPT system With dual NEMR structure at 100 mm. **c** Primary current versus secondary voltage of WPT system without NEMR structure at 150 mm. **d** Primary current versus secondary voltage of WPT system With dual NEMR structure at 150 mm. **e** Primary current versus secondary voltage of WPT system without NEMR structure at 200 mm. **f** Primary current versus secondary voltage of WPT system With dual NEMR structure at 200 mm. **g** Mutual inductance versus transfer distance of the WPT systems with and without NEMR structures.

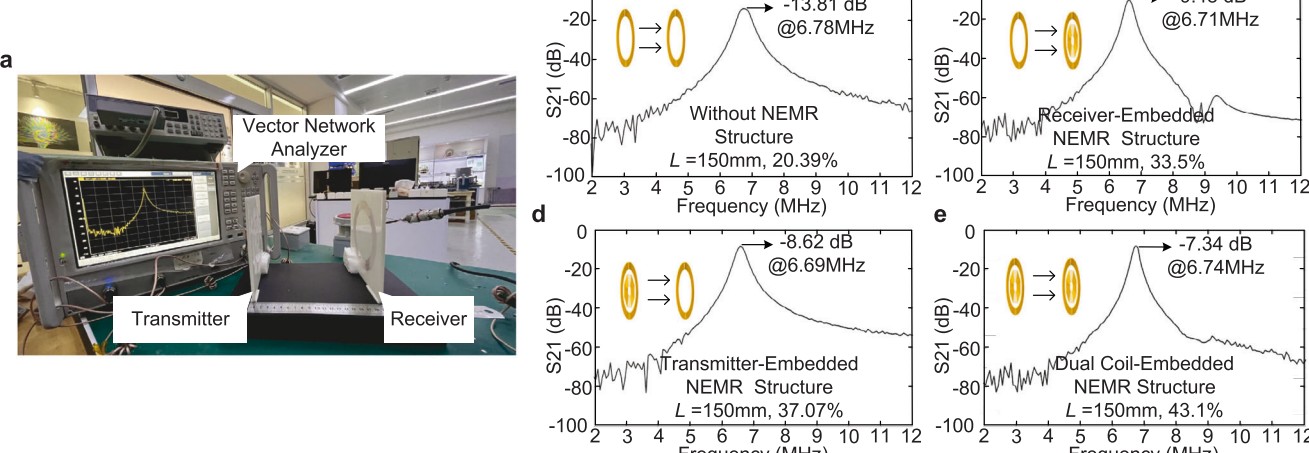

**Fig. 5 | Transmission gain experiments and results. a** Experimental platform for transmission gain of the NEMR structure-based WPT system. **b** Transmission gain under 150 mm without NEMR structure. **c** Transmission gain under 150 mm with transmitter-embedded NEMR structure. **d** Transmission gain under 150 mm with receiver-embedded NEMR structure. **e** Transmission gain under 150 mm with dual-embedded NEMR structure.

The weak-coupling WPT system with dual coil-embedded NEMR structure has the highest transmission gain among the above-mentioned systems, reaching 43.1% under the transfer distance of 150 mm, see Fig. 5b–e. The results also indicate that the NEMR structure with different installation positions can all increase the efficiency of the WPT system, while the efficiency enhancing effects vary from its installation positions and quantity. Compared to the WPT system without the NEMR structure, the efficiency promotions with dual coil-embedded, transmitter-embedded, and receiver-embedded NEMR structures are 111.3%, 82.2%, and 65.1%, respectively.

**Power transfer efficiency measurement**
To further verify the effectiveness of the proposed NEMR structure in efficiency enhancement for the weak coupling WPT system, power experiments are conducted. The verification system consists of a signal generator (TG5011), a power amplifier (ATA-1222A), and a four-channel oscilloscope, which is indicated in Fig. 6a. The electric parameters of the coils and NEMR structure is given in Table 1. The power transfer efficiency $\eta$ of the WPT system is the proportion between the output power and input power, expressed as.

$$\eta = \frac{P_{out}}{P_{in}} = \frac{I_s^2 R_L}{U_p I_p \cos\theta_s} \tag{4}$$

where $U_p$ and $I_p$, are the voltage and current of the transmitter coil. $\theta_s$ is the phase difference between $U_p$ and $I_p$. $I_s$ and $R_L$ are the current in the receiver coil and the load resistance, respectively.

The experiment results of the WPT systems with and without the proposed dual NEMR structures are given in Fig. 6b, c. The measured waveforms in channels 1-4 are the input voltage, input current, voltage of the receiver coil and output current. As shown in Fig. 6b, c, the NEMR structure increases the current of the receiver coil and, accordingly, enhances the transfer efficiency of the weak coupling WPT system. Based on the experiment results of the transmission gain and power transfer efficiency, the performance of the WPT system with and without the proposed NEMR structure under different transfer distances is given in Fig. 6d. Significantly, the power experiments under different transfer distances are conducted with a constant input voltage.

The efficiency and transmission gain of the systems versus angular misalignments are given in Fig. 6e, where the misalignment is from 0 to 50 degrees. As demonstrated in Fig. 6e, the dual coil-embedded NEMR structure can effectively enhance the efficiency and the transmission gain compared to the system without the NEMR structure. The parallel condition is given in Fig. 6f. The efficiency promotion caused by the dual coil-embedded NEMR structure is from 97.3 to 106.15% compared to the system without the NEMR structure under the parallel misalignment from 20 to 100 mm. The verification results indicate that the

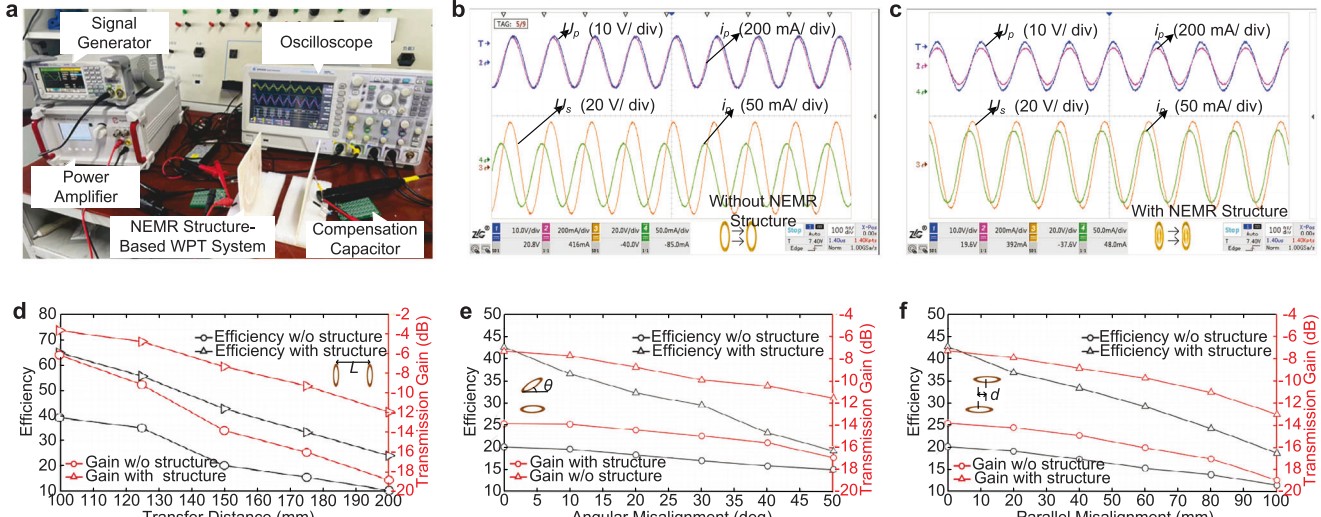

**Fig. 6 | Power experiments and results. a** Experimental platform for the power experiments of the NEMR structure-based WPT system. **b** Power experiment of the system without the dual NEMR structure under the transfer distance of 150 mm. **c** Power experiment of the system with the dual NEMR structure under the transfer distance of 150 mm. **d** Efficiency and transmission gain of the WPT systems with/ without the proposed NEMR structure versus transfer distance. **e** Efficiency and transmission gain of the WPT systems with/without the NEMR structure under angular misalignment under the transfer distance of 150 mm. **f** Efficiency and transmission gain of the WPT systems with/without the NEMR structure under parallel misalignment under the transfer distance of 150 mm.

proposed NEMR structure can effectively enhance the efficiency and transmission gain under parallel misalignment conditions.

To conduct a detailed comparison with the previously proposed solutions, Table S4 is given (see Supplementary Note 7). The proposed design has a good efficiency enhancement capability among previously proposed solutions. In essence, the proposed design requires no additional space for structure installation, which confirms its practicability in WPT systems with a long transfer distance or small coils. The loss caused by the NEMR structure shares only 6.92% compared to the total loss of the system, which is not a serious burden of the WPT system (see Supplementary Note 8 for details).

## Discussion

In this work, the concept of a negative equivalent magnetic reluctance (NEMR) structure and its modelling method, as well as its application in a weak coupling WPT system are presented. The contributions of this paper are concluded as follows. (i) The modelling method of the NEMR structure based on $L/C$ parameters and magnetic reluctance is proposed, extending the metamaterial theory based on Snell's law. (ii) The negative equivalent magnetic reluctance property is verified via the transformer experiments with the core of the stacked NEMR structure. (iii) The mutual inductance enhancement property of the NEMR structure in the weak coupling WPT system is verified, which can increase the mutual inductance by more than 154.2%. (iv) The effectiveness of the designed NEMR structure is validated based on efficiency comparison, Incorporated into the designed topology, the transfer efficiency enhancement of the aforementioned system is from 33.4 to 121.9% under different transfer distances. (vi) The NEMR structure presents a low loss, occupying only 6.72% compared to the

total loss of the system. Based on the abovementioned results, the effectiveness of the modelling method and the NEMR structure on efficiency enhancement of the WPT system are verified.

## Method

### NEMR structure-based WPT system configuration

The equivalent magnetic circuit of the proposed system is given, see Fig. 7a. The magnetomotive force $F_T$ is generated by the current in the transmitter coil. The flux $\phi_p$ is the total flux generated by the transmitter coil which consists of the mutual flux $\phi_m$ and the leakage flux $\phi_l$. The relationship between those fluxes based on Fig. 7a is given as.

$$\phi_p = \phi_m + \phi_l = \underbrace{\phi_{m1} + \phi_{ml}}_{Mutual\,flux} + \underbrace{\phi_{l1} + \phi_{ll}}_{leakage\,flux} \quad (5)$$

As for the magnetic reluctance, the $R_{l1}$, and $R_{l2}$ are used to represent that the magnetic reluctance in the branches is composed of the parallel combination of the magnetic reluctances, considering the closed magnetic field consisted of numbers of the magnetic flux lines. $R_{core}$ is the equivalent magnetic reluctance of the core material. The self-inductance and mutual-inductance of the WPT coil is determined by the magnetic reluctance of the magnetic circuit, defined as

$$\begin{bmatrix} L \\ M \end{bmatrix} = N^2 \begin{bmatrix} \underbrace{[R_{Core}+R_i]^{-1}}_{Leakage\,Inductance} + \underbrace{[2R_{Core}+R_j]^{-1}}_{Mutual\,Inductance} \\ [2R_{Core}+R_j]^{-1} \end{bmatrix} \quad (6)$$

where $R_i$ equals $[(R_{l1}+R_{l3})\,(R_{m2}+R_{l4})]/(R_{m2}+R_{l4}+R_{l2}+R_{l3})$ and $R_j$ equals $[(R_{m1}+R_{l3})\,(R_{m2}+R_{l4})]/(R_{m2}+R_{l4}+R_{m1}+R_{l3})$. $R_{core}$ is the magnetic reluctance of the magnetic core determined by the material. (The detailed analysis is given in Supplementary Note 9).

The mutual inductance of the system with NEMR structure is inherently larger than that with the core material of ferrite or air, considering the negative magnetic reluctance. The generalized equivalent circuit of the WPT system with the series-series

## Table 1 | Electric parameters of the WPT coils and NEMR structure

| | Transmitter/Receiver | NEMR structure |
|---|---|---|
| Self-inductance(uH) | 17.2 | 9.4 |
| Resistance (ohm) | 2.31 | 1.12 |
| Compensation Capacitor (pF) | 33 | 108 |

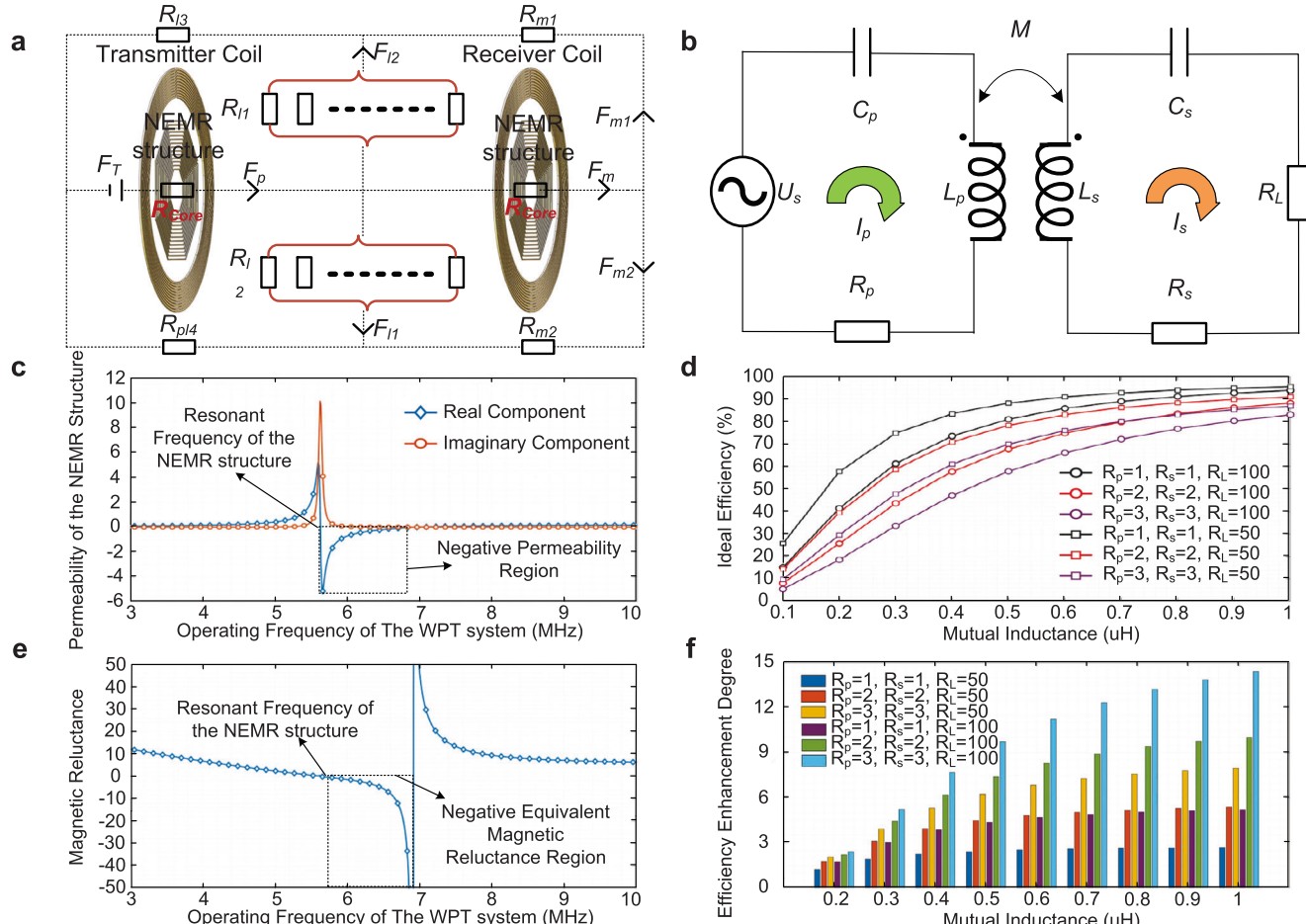

**Fig. 7 | Modelling of NEMR structure and its application in weak-coupling WPT system configuration. a** Experimental platform for the power experiments of the NEMR structure-based WPT system. **b** Power experiment of the system without the dual NEMR structure under the transfer distance of 150 mm. **c** Power experiment of the system with the dual NEMR structure under the transfer distance of 150 mm. **d** Efficiency and transmission gain of the weak-coupling WPT systems with/without

the proposed NEMR structure versus transfer distance. **e** Efficiency and transmission gain of the WPT systems with/without the NEMR structure under angular misalignment under the transfer distance of 150 mm. **f** Efficiency and transmission gain of the WPT systems with/without the NEMR structure under parallel misalignment under the transfer distance of 150 mm.

compensation network is employed and shown in Fig. 7b. The efficiency $\eta$ of the weak-coupling WPT system[35] is expressed as follows (The detailed analysis is given in Supplementary Note 5).

$$\eta = \frac{P_{out}}{P_{in}} = \frac{1}{1 + \frac{R_s}{R_L} + \frac{R_p}{R_L}\left[\frac{(R_s + R_L)}{\omega M}\right]^2} \quad (7)$$

where $R_p$ and $R_s$ are the resistors of the transmitter coil and receiver coil.

As for the generalized kilohertz WPT system with a short transfer distance[36,37] (smaller than a quarter of coil diameter) and large coil size, the mutual inductance $M$ between the primary and secondary coil is larger enough (about mH) and the coupling coefficient is always larger than 0.15. In this condition, $R_p$ and $R_s$ would have a slight impact on the efficiency. However, as for the weak-coupling WPT system, the mutual inductance and coupling coefficient are small. Therefore, the efficiency of the system is highly dependent on the resistors $R_p$ and $R_s$. Based on Eq. (7), the mutual inductance $M$ versus the transfer efficiency of the weak coupling WPT system under 6.78 MHz with different resistance parameters are given in Fig. 7d. Results indicate that increasing the mutual inductance in weak-coupling WPT systems can significantly enhance the transfer efficiency, see Fig. 7f.

## Modelling of NEMR structure

The electric circuit-based analysis applied to the NEMR structure[38] is employed, to indicate the feasibility of the negative permeability under the designed frequency range, defined as.

$$\mu_r = 1 + \frac{\mu_0}{LV}\frac{\omega^2}{\omega_0^2 - \omega^2 + j\frac{R\omega}{L}}\sum_{k=1}^{N} s_k^2 \quad (8)$$

where $L$ and $V$ are the effective inductance and volume of the structure. $\omega_0$ and $\omega$ are the resonant frequency of the structure and operating frequency of the system. $R$ and $s_k$ are the resistor and sectional area of the NEMR structure.

The permeability $\mu_r$ and magnetic reluctance of the NEMR structure $R_{NEMR}$ can be obtained based on Eq. (8), defined as

$$R_{NEMR} = \frac{l}{\mu A} = \frac{l}{A\left(1 + \frac{\mu_0}{LV}\frac{\omega^2}{\omega_0^2 - \omega^2 + j\frac{R\omega}{L}}\sum s_k^2\right)} \quad (9)$$

where $A$ is the cross-sectional area of the circuit in square meters. (The detailed formula derivation see Supplementary Note 10)

Based on Eqs. (8), (9), the relationship between the operating frequency and the permeability/magnetic reluctance in this design is indicated in Fig. 7c, e. In the weak-coupling WPT system, the

decreased magnetic reluctance refers to increased mutual inductance, see Eq. (7). By modifying the compensation capacitor $C$, the resonance frequency of the system $\omega_0$ will be varied, and accordingly, the NEMR structure can indicate a negative equivalent magnetic reluctance and the efficiency of the weak-coupling WPT system will be enhanced.

## Data availability
The data that support the findings of this study are presented in Supplementary Information. The source data underlying Figs. 3k, 4g, 5b–e, 6d–f and 7c–f are provided in the Source Data files with this paper or available from the corresponding author on request. Source data are provided with this paper.

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

## Acknowledgements

This work was supported by the Hong Kong Research Grants Council Collaborative Research Fund, Hong Kong Special Administrative Region, China, under Grant C1052-21G.

## Author contributions

Y.C., W.F. and S.N. conceived the idea and conducted the theoretical analysis. Y.C. performed the simulation and experiments. Y.C., S.N. and W.F. wrote the manuscript, analyzed the data and interpreted the results. H.L. provided the suggestion and discussion. Funding was acquired by S.N. This work was supervised by S.N. and W.F. All authors reviewed and edited the manuscript.

## Competing interests

The authors declare no competing interests.
