## [Peer Review File · Nature Communications]

Modelling of negative equivalent magnetic reluctance structure and its application in weak-coupling wireless power transmissionREVIEWER COMMENTS

Reviewer #1 (Remarks to the Author):

The title is interesting, although the advantages of using the NEMR circuit have not been justified.

Wireless Power Transfer (WPT) systems are typically designed at the critical coupling case to achieve maximum efficiency and need improvement in performance at weak coupling without compromising efficiency at critical coupling.

From the authors' analysis, it is not clear how the NEMR affects efficiency at critical coupling, i.e., the maximum efficiency.

Lastly, the comparison is not fair. They should reference well-designed WPT systems, especially those using MM (metamaterials). A few examples are:

1. M. Aboualalaa et al, "Reliable Multiple Cascaded Resonators WPT System Using Stacked Split-Ring Metamaterial Passive Relays," IEEE Transactions on Instrumentation and Measurement, vol. 72, paper no. 8006710, DOI: 10.1109/TIM.2023.3324672.
2. Y. Ikeda et al, "Stacked metasurfaces for misalignment improvement of WPT system using spiral resonators," the 51st European Microwave Conference, pp. 257-260, DOI: 10.23919/EuMC50147.2022.9784281.

Please ensure that the advantages of the NEMR circuit are clearly outlined and address the impact on efficiency at critical coupling in your analysis. Additionally, make a fair and comprehensive comparison by referencing relevant, well-designed WPT systems, especially those utilizing metamaterials.

Reviewer #2 (Remarks to the Author):

General comments:

- The manuscript is based on a magnetic circuit model of a metamaterial embedded with transmitter and receiver coils in a Wireless Power Transfer system for calculating the reluctance and mutual inductance of the metamaterial. Unfortunately, the magnetic circuit model, from my point of view, is not well-described and explained. Furthermore, the experimental verification is not clear.

The overall presentation of the work confuses the reader. The authors mention a stacked version of the metamaterial, in the abstract, but finally they do not use it for experimental verification.

-I suggest a global revision of the English language.

Comments and remarks:

- The following sentence is not clear: "However, considering that the hysteresis loss of generalized iron-oxide ferrite will boost if the operating frequency of the WPT system is higher than the rated

frequency" I think some verbs of conclusions are missing.

- The following sentences are not very clear about what they refer to. Please, revise accordingly.

"However, those three types of solutions both have some inherent drawbacks. First, the hysteresis loss of the

magnetic ferrite core will boost with the increase in operating frequency." Probably, some conclusions are missing.

-Page 2: "increases" instead "increase"?

- This sentence "This issue results that the high operating frequency and magnetic ferrite core can not be applied in weak coupling WPT systems simultaneously." I think is redundant.

- The equivalent magnetic circuit of the system is not clear. The authors define five reluctances parameters R_{p0} , R_{p1} , R_{p2} , R_{p3} and R_{p4} , which they describe as the magnetic reluctance of the

branches of the transmitter coil. What the branches of the transmitter coil are? Why the reluctances parameters are in a number of five? Please, revise and explain accordingly. A same clarification is required for parameters R_{m0} , R_{m1} and R_{m2} and R_{core} .

- In table II, the authors define a compensation capacitor, which is the first time they introduce in the manuscript. Please, could you explain in which part of their system they are using a compensation capacitor? Probably the equivalent electrical circuit of the system needs to be presented before.

- Equation (4): the value assumed from the reluctance in air I think is not correct. Instead of μ_h , you probably mean " μ_{h0} "?

-Page 7: " R_{fer} " instead of " R_{ref} "?

- Equation (6): Definition for $P_{in} = U_s I_p$ is not precise from electrical circuit theory point of view. Please revise considering that the equation are referred to an AC circuit.

-Equation (8): for defining the efficiency, the authors cite a manuscript, [22]. Actually, in [22] it seems that there is not a definition of the efficiency. Please, revise accordingly.

- The authors selected the operational frequency equal to 6.78MHz. Please, could you explain why they choose this frequency?

_ Equation (9) is completely uncorrect from conceptual point of view. The authors mixed a treatment in phasorial regime and temporal regime. Please revise accordingly.

- It is not clear the design of the system. What the value for parameter n is?. Please clarify this aspect.

- Equation (14): What the parameter A is in the equation? Please, explain.

- Electromagnetic analysis section: please give more information about the numerical analysis you performed. Which software used, the EM meshgrid, and so the number of unknowns, the hardware performance they used for simulations.

- Fig.6: explaining the results of figure 6, the authors say that "As shown in Fig. 6, the WPT system with the dual coil-embedded NEMR structure has the best performance,". I actually do not understand what is the difference between the field maps presented in Fig. 6. They look very similar each others. Please, clarify this aspect and I suggest to use a colormap scale which highlights the differences, if there are. What about the simulated efficiency of the system? Is it comparable with the results of experimental analysis?

-Fig.8: "source" instead of "sourse"

-Equation (16): the sentence "equal the corresponding magnetic reluctance excepting that of core material in Eq. (3)" has not sense.

- The experimental analysis is a little bit confusing the reader. The authors examine a stacked version of the NEMR structure, mentioned only in the abstract. Please, clarify this aspect. Did the authors perform electromagnetic simulation of the stacked version of the NEMR structure? Could you explain why do you use a stacked version of NEMR? What is the distance between each layer? Please, give more details.

- the transmission gain verification is performed with a stacked NEMR structure or a single NEMR? From figures, It looks like that the verification has been performed with a not stacked version of NEMR. Why? Please, clarify this aspect.

- Conclusions section: the value for mutual inductance improvement is not readable.

Dear Reviewers of Nature Communication,

Original ID NCOMMS-23-52178 entitled 'Modelling of Negative Equivalent Magnetic Reluctance Structure and Its Application in Weak-Coupling WPT Systems'.

Thank you for your comments on our manuscript. We have revised our manuscript according to your comments and suggestions. All the specific suggestions and remarks provided by the reviewers have been incorporated in the revised version of the manuscript. The changes in the revised manuscript are highlighted in **red color** in this new version of the manuscript.

In summary, the following changes are applied to the submitted manuscript:

- 1) Based on the comments, Section II, Section III, and Section V have been revised by removing some inaccurate descriptions and mistakes.
- 2) The description of the main contribution of this manuscript has been revised.
- 3) The analysis of the magnetic reluctance of the WPT system with NEMR structure has been reorganized.
- 4) More well-designed metamaterial-based WPT systems have been supplemented for comparison and the results are given in Table 4 of the revised manuscript.
- 5) The description and the purpose of the design of the single layer NEMR structure and stacked NEMR structure have been reorganized.
- 6) The simulation information and conditions have been supplemented in the revised manuscript.
- 7) The analysis of the transformer with the stacked NEMR structure is reorganized.
- 8) The introduction and conclusion are carefully revised based on the reviewer's comments.

Response to Reviewers:

Reviewer 1:

The authors would like to thank Reviewer 1 for the constructive comments and suggestions and we would like to provide the following explanations for the comments. The changes in the revised manuscript are highlighted in red colour.

Comment-1. The title is interesting, although the advantages of using the NEMR circuit have not been justified (SubQ1). Wireless Power Transfer (WPT) systems are typically designed at the critical coupling case to achieve maximum efficiency and need improvement in performance at weak coupling without compromising efficiency at critical coupling. (SubQ2)

Response-1: Thank you for your comment. To better illustrate this issue, **Comment-1** is divided into two separate questions.

SubQ1. This paper is not only focused on employing the negative equivalent magnetic reluctance (NEMR) structure to increase the efficiency of wireless power transfer (WPT) systems but also proposes and investigates the model of negative equivalent magnetic reluctance based on metamaterial theory. The metamaterial theory relies on Snell's law to refract or reflect electromagnetic waves with negative/zero permeability [R1]-[R4]. The basic schematic diagram of the effect of metamaterial on the WPT system is given in Fig. R1.

Fig. R1 The effect of metamaterial on wireless power transfer system based on Snell's Law. (a) Metamaterial with refraction property [R1]-[R2] (permeability $\mu < 0$). (b) Metamaterial with reflection property [R3]-[R4] (permeability $\mu \approx 0$)

As shown in Fig. R1, the metamaterial can refract or reflect the electromagnetic wave, increasing the flux on the receiver coil. The increased flux can increase the induced voltage and, accordingly, increase the transfer efficiency.

According to traditional metamaterial theory, installing metamaterial in the receiver coil cannot refract or reflect electromagnetic waves, thereby failing to enhance transfer efficiency. However, contrary to conventional theory, incorporating metamaterial in the receiver coil effectively boosts efficiency, which cannot be properly explained by conventional metamaterial theory.

The advantages of this NEMR structure are summarized as follows. i) the dual-coil embedded structure can further increase the efficiency of the system compared to the conventional metamaterial-based WPT system; ii) the proposed magnetic modelling method can explain the effectiveness of the NEMR structure installed in the receiver coil, which cannot be achieved by the metamaterial theory. iii) The NEMR structure exhibits high versatility and occupies no additional space for slab installation except in transmitter and receiver coils; iv) the design of the NEMR

structure provides a high quality factor compared to the generalized metamaterial slab consisting of many metamaterial units, which has a high operating frequency range considering the loss.

SubQ2. Thank you for your comment. Ideally, the efficiency will be stable under the critical coupling condition if the product of the operating frequency ω and the mutual inductance M is large enough. However, the resistance of the coil will greatly influence the efficiency when ωM is not large enough.

To clearly illustrate the impact of parameters on system efficiency, the efficiency calculation incorporating mutual inductance, operating frequency and resistor is given as follows. Taking the example of a Series-Series compensation topology in a weak-coupling WPT system, the transmission efficiency under the fully resonance condition is expressed as in Eq. (R1).

$$\eta = \frac{P_{out}}{P_{in}} = \frac{1}{1 + \frac{R_s}{R_L} + \frac{R_p}{R_L} \left[\frac{(R_s + R_L)}{\omega M} \right]^2} \quad (R1)$$

As shown in Eq. (R1), under the condition of full resonance and constant impedance, the transmission efficiency of the WPT system is directly proportional to the operating frequency and mutual inductance. However, when the operating frequency is too low to maintain the efficiency with weakly coupled conditions (low mutual inductance), increasing the mutual inductance can effectively enhance the efficiency. To verify this point, the relationships between mutual inductance M and system efficiency as well as the efficiency enhancement degree under the frequency of 6.78 MHz are given in Fig. R2.

Fig. R2 The impact of mutual inductance with different parameters on efficiency under 6.78 MHz. (a) Ideal efficiency versus mutual inductance. (b) Efficiency enhancement degree compared to the system with the mutual inductance of 0.1 μ H.

As for the design of mutual inductance M , the area of coils and the transfer distance between the coils need to be mutually matched with the application scenario. The relationship among the size of coils and transfer distance between coils determine the mutual inductance M as well as coupling coefficient k , is expressed as follows.

$$M = \frac{\phi}{I} \approx \frac{\mu N_1 N_2 A}{d} \quad (R2)$$

$$k = \frac{M}{\sqrt{L_1 L_2}} \quad (R3)$$

where A is the overlap area of the coils; μ is the permeability of the transmission medium; N_1, N_2 are the number of turns of the coils; d is the transmission distance between coils. L_1, L_2 are the self inductance of the primary and secondary coils.

Assuming that the two coils of the WPT system are concentric, the mutual inductance is inversely proportional to the transmission distance d .

As for the resistance of the coil, the skin effect under different frequencies and the design considerations that should be taken into consideration are addressed as follows.

The operating frequency of the WPT system greatly affects the efficiency of the WPT system. However, the operating frequency of the WPT system should consider the skin effect of the WPT coil material and the operating frequency of the potential power devices.

For the skin effect, the copper wire diameter should be less than twice of the skin depth d_s at the operational frequency, expressed as follows.

$$d_s = \sqrt{\frac{1}{\pi f \phi \mu_s}} \tag{R4}$$

where ϕ is the electrical conductivity of the copper wire; μ_s is the magnetic permeability of the copper wire; f is the operating frequency.

The relationships of skin depth and maximum coil diameter with frequency variations are given in Fig. R3.

Fig. R3 The relationships between the maximum diameter of the coil, skin depth and frequency.

Based on the results in Fig. R3, the 20-turn Liz wire with a wire diameter of 0.05 mm is chosen for both transmitter and receiver coils. This design aims to mitigate the impact of skin effect on the wire resistance.

As for the design of operating frequency, considering the potential high-frequency region, GaN devices are commonly used in the MHz region, with a maximum operating frequency which can reach up to 10 MHz.

Based on the above mentioned considerations of WPT system design, the operating frequency is selected as 6.78 MHz considering the Resence standard and the resistor of the coil is reduced with the employment of Liz wire.

Based on the results given in Fig. R2, under a constant operating frequency, increasing the mutual inductance can effectively increase the efficiency of the weak coupling WPT system. To increase the efficiency based on the above mentioned characteristics, we propose a negative magnetic reluctance structure to reduce the magnetic reluctance, thereby increasing the mutual inductance and efficiency of weakly coupled WPT systems.

[R1]M. Aboualalaa and R. K. Pokharel, "Reliable Multiple Cascaded Resonators WPT System Using Stacked Split-Ring Metamaterial Passive Relays," *IEEE Transactions on Instrumentation and Measurement*, vol. 72, pp. 1-10, 2023, Art no. 8006710, doi: 10.1109/TIM.2023.3324672.

[R2]Y. Cho et al., "Thin Hybrid Metamaterial Slab With Negative and Zero Permeability for High Efficiency and Low Electromagnetic Field in Wireless Power Transfer Systems," in IEEE Transactions on Electromagnetic Compatibility, vol. 60, no. 4, pp. 1001-1009, Aug. 2018, doi: 10.1109/TEMC.2017.2751595.

[R3]C. Lu, X. Huang, X. Tao, C. Rong and M. Liu, "Comprehensive Analysis of Side-Placed Metamaterials in Wireless Power Transfer System," IEEE Access., vol. 8, pp. 152900-152908, 2020.doi: 10.1109/ACCESS.2020.3017492.

[R4] C. Lu et al, "Investigation of Negative and Near-Zero Permeability Metamaterials for Increased Efficiency and Reduced Electromagnetic Field Leakage in a Wireless Power Transfer System," IEEE Trans Electromagn Compatibility., vol. 61, no. 5, pp. 1438-1446, Oct. 2019.

Changes Made: i) Fig. R2(b) is supplemented in the revised manuscript as Fig. 4(b). ii) The main contribution of this work is rewritten in the section of INTRODUCTION of the revised manuscript.

Comment-2. From the authors' analysis, it is not clear how the NEMR affects efficiency at critical coupling, i.e., the maximum efficiency.

Response-2: The relationship between the operating frequency, mutual inductance/coupling coefficient and efficiency is concluded in **SubQ2 of Response-1**. The impact and analysis of NEMR structure on mutual inductance and efficiency are reorganized as follows.

The equivalent magnetic circuit of the NEMR structure-based WPT system is revised from Fig. R4(a)to Fig. R4(b).

Fig. R4. The equivalent magnetic circuit of the proposed design. (a) Original magnetic circuit analysis. (b) Revised magnetic circuit analysis

For a two-coil system with the same number of turns of coils, the self-inductance L , leakage inductance L_l , and mutual inductance M should follow Eq. (R5).

$$L = L_l + M = \frac{\psi}{i} = \frac{N^2}{R_m} \quad (R5)$$

where ψ is the flux linkage of the coil and i is the current through the coil.

For a two-coil system with the same number of turns, based on Eq. (R5), the relationships among the magnetic reluctance, the number of turns of coils, flux, and the current through the coil are concluded as follows.

$$\begin{bmatrix} L \\ M \end{bmatrix} = \begin{bmatrix} L_l + M \\ M \end{bmatrix} = \begin{bmatrix} \psi_l \\ \psi_m \end{bmatrix} [i^{-1}] = \begin{bmatrix} N(\phi_l + \phi_m) \\ N\phi_m \end{bmatrix} [i^{-1}] \quad (R6)$$

where ψ_l and ψ_m are the leakage flux linkage and main flux linkage, respectively; while ϕ_l and ϕ_m are the magnetic flux related to leakage inductance and mutual inductance, respectively.

Based on Eq. (R5) and Fig. R4, Eq. (R6) can be rewritten as Eq. (R7).

$$\begin{bmatrix} L \\ M \end{bmatrix} = \begin{bmatrix} L_l + M \\ M \end{bmatrix} N^2 \begin{bmatrix} \underbrace{\left[R_{Core} + \frac{(R_{l1} + R_{l3})(R_{l2} + R_{l4})}{R_{l2} + R_{l4} + R_{l2} + R_{l3}} \right]^{-1}}_{\text{Leakage Inductance}} + \underbrace{\left[2R_{Core} + \frac{(R_{m1} + R_{l3})(R_{m2} + R_{l4})}{R_{m2} + R_{l4} + R_{m1} + R_{l3}} \right]^{-1}}_{\text{Mutual Inductance}} \\ \left[2R_{Core} + \frac{(R_{m1} + R_{l3})(R_{m2} + R_{l4})}{R_{m2} + R_{l4} + R_{m1} + R_{l3}} \right]^{-1} \end{bmatrix} \quad (R7)$$

where R_{Core} is the magnetic reluctance of the magnetic core determined by the material.

Based on Eq. (R7) and the results in Fig. R2, the decreased R_{Core} can increase the mutual inductance M and the efficiency of the weakly coupled WPT system.

If the core of the system adopts the NEMR structure, R_{Core} can be expressed as follows.

$$R_{Core} = R_{NEMR} = \frac{l}{A \left(1 + \frac{\mu_0}{LV} \frac{\omega^2}{\omega_0^2 - \omega^2 + j\frac{R\omega}{L}} \sum_{k=1}^{N=14} S_k^2 \right)} \quad (R8)$$

The relationship between the operating NEMR frequency and the permeability/magnetic reluctance of NEMR structure based on Eq. (R8) is indicated in Fig. R5.

Fig. R5. The relationships among the permeability/magnetic reluctance of NEMR structure and operating frequency. (a) permeability. (b) magnetic reluctance.

As indicated in Fig. R5, the permeability of the NEMR structure is negative under the negative equivalent magnetic reluctance region. Under this region, based on Eq. (R7) and the results in Fig. R2, the NEMR structure can effectively increase the mutual inductance and efficiency of the weakly coupled WPT system.

To clearly illustrate this trend, the impact of NEMR structure on mutual inductance (coupling coefficient) under different transfer distances and the enhancement degree is given in Fig. R6.

Fig. R6. The impact of NEMR structure on mutual inductance (coupling coefficient). (a) Mutual inductance (coupling coefficient) versus transfer distance. (b) Mutual inductance (coupling coefficient) enhancement degree versus transfer distance.

Changes Made: Eqs. (R5), (R6), and (R7) as well as the corresponding description have been supplemented in the revised manuscript as Eqs. (1), (3), (4).

Comment-3. Lastly, the comparison is not fair. They should reference well-designed WPT systems, especially those using MM (metamaterials). A few examples are:

[1] M. Aboualalaa et al, "Reliable Multiple Cascaded Resonators WPT System Using Stacked Split-Ring Metamaterial Passive Relays," IEEE Transactions on Instrumentation and Measurement, vol. 72, paper no. 8006710, DOI: 10.1109/TIM.2023.3324672.

[2] Y. Ikeda et al, "Stacked metasurfaces for misalignment improvement of WPT system using spiral resonators," the 51st European Microwave Conference, pp. 257-260, DOI: 10.23919/EuMC50147.2022.9784281.

Please ensure that the advantages of the NEMR circuit are clearly outlined and address the impact on efficiency at critical coupling in your analysis.(SubQ1) Additionally, make a fair and comprehensive comparison by referencing relevant, well-designed WPT systems, especially those utilizing metamaterials. (SubQ2)

Response-3: Thank you for your comment. To better illustrate this issue, **Comment-3** is divided into two separate questions.

SubQ1. The advantages of the NEMR structure and the analysis method in this manuscript are concluded as follows.

i) The magnetic reluctance-based modelling method is established to enhance the metamaterial theory, extending the negative permeability effect to the negative magnetic reluctance effect on the multi-coil system.

ii) The NEMR structure modelling method properly explains how the resonators installed in the receiver coil, which cannot reflect or refract the magnetic field, increase the mutual inductance and efficiency of the weak coupling WPT systems.

iii) Compared to conventional metamaterial design, the quality factor of NEMR structure is increased with similar size of metamaterial slab consisting of metamaterial unit. The property allows the NEMR structure to be used in the WPT system with a lower frequency (low frequency will decrease the quality factor and increase the loss of coil).

iv) The NEMR structure occupies no additional space for installing the metamaterial/NEMR structure slab, which makes it suitable for most of the WPT applications.

To clearly illustrate the advantages of the proposed NEMR structure-based WPT system compared to the system with metamaterial/metasurface/resonance coil, a comprehensive comparison is given in Table R1.

Table R1. General Comparison of the Proposed NEMR Structure and State-of-art Works for Weak-Coupling WPT Systems

Categorization	No extra space occupying	Wide frequency range	No unconventional material required	High quality factor
This paper. NEMR structure	√	√	√	√
[R5] Ferrite	√	×	√	-
[R6]-[R7] Designed magnetic core	√	×	×	-
[R8]-[R9] Resonance coil	×	√	×	√
[R10] Superconductivity coil	×	√	√	√
[R11] Metamaterial	×	×	√	×
[R12] Metasurface	√	×	√	×

As for the impact of NEMR structure of efficiency at critical coupling, a detailed analysis is given in Response-2, which analyzes the impact of mutual inductance, resistor and operating frequency on efficiency.

SuQ2. Thank you for your suggestion, the comparison of metamaterial-based WPT systems is supplemented and given in Table R2. The comparison parameters include the operating frequency, volume proportion between resonator and transmitter/receiver coil, rated transfer distance, and efficiency with/without the resonator. In this comparison, the metamaterial, metasurface, relay resonator, spiral resonator and repeater coil are described as resonators.

Table R2. Comprehensive Comparison Between Previously Reported Solutions and The Proposed Design.

Category	Operating Frequency (MHz)	Diameter of Tx/Rx Coils (mm)	Volume proportion of Resonator and Tx/Rx	Rated Transfer Distance (mm)	Efficiency without/with resonator (%)
This paper. NEMR structure	6.78	120	0.625	150	20.9/41.2
[R11]. Single Metamaterial	6.78	150	1.73	150	34.5/41.7
[R13]. Cubic metamaterial	560	30	1.23	100	10.2/17
[R14]. Metasurface	430	40/10	1.33/0.33	60	0.507/4.09
[R15]. Tunable metamaterial	6.78	600	1.25	900	27.2/63.4
[R16]. Cascaded metamaterial	71	30	1.039	80	50/73
[R12]. Tx-metasurface	13.56	124	0.758	200	37.7/42.2
[R17]. Relay resonator	6.78	300	1.142	350	45/64
[R18]. Spiral resonator	6.78	350	1.142	400	36.59/57.2
[R19]. RepeaterCoil	13.56	90/80	1.16/1.32	56	45.1/77.2

Ref. [1] is expressed as Ref. [R16] in table R1. Considering Ref. [2] focuses on improving the misalignment tolerance of the WPT system with metamaterial, the comparison may be unfair with the existing solutions aiming to enhance efficiency. Hence, well-designed WPT systems [R11]-[R19] are supplemented for conducting a fair and comprehensive comparison.

- [R5] H. Wang, K. W. E. Cheng and Y. Yang. A New Resonator Design for Wireless Battery Charging Systems of Electric Bicycles. *IEEE Trans. Emerg. Sel Power Electron.*, vol. 10, no. 5, pp. 6009-6019, Oct. 2022. <https://doi.org/10.1109/JESTPE.2022.3157729>.
- [R6] T. Ide, N. Imaoka, K. Ozaki, M. Shimizu and N. Takada. Ndx Fe1-x Ny Magnetic Core Application for Resonance Coil of 13.56 MHz GaN Wireless Power Transmission. *IEEE Trans. Magn.*, vol. 55, no. 10, pp. 1-5, Oct. 2019. <https://doi.org/10.1109/TMAG.2019.2925054>.
- [R7] D. Miura, Y. Tokudaiji, K. Murasato, Y. Hattori, Y. Bu and T. Mizuno, "Investigation of Structure and Material for Back Yoke at 13.56 MHz Wireless Power Transfer Focused on High Transmission Efficiency," *IEEE Trans. Magn.*, vol. 55, no. 7, pp. 1-5, July 2019. <https://doi.org/10.1109/TMAG.2019.2895199>.
- [R8] D. Miura, Y. Tokudaiji, K. Murasato, Y. Hattori, Y. Bu and T. Mizuno, "Investigation of Structure and Material for Back Yoke at 13.56 MHz Wireless Power Transfer Focused on High Transmission Efficiency," *IEEE Trans. Magn.*, vol. 55, no. 7, pp. 1-5, July 2019. <https://doi.org/10.1109/TMAG.2019.2895199>.
- [R9] X. Liu and G. Wang. A Novel Wireless Power Transfer System With Double Intermediate Resonant Coils. *IEEE Trans. Ind. Electron.*, vol. 63, no. 4, pp. 2174-2180, April 2016, <https://doi.org/10.1109/TIE.2015.2510512>.
- [R10] N. Oshimoto, K. Sakuma and N. Sekiya, Improvement in Power Transmission Efficiency of Wireless Power Transfer System Using Superconducting Intermediate Coil, *IEEE Trans. Applied Superconductivity*, vol. 33, no. 5, pp. 1-4, Aug. 2023. <https://doi.org/10.1109/TASC.2023.3256342>.
- [R11] Y. Cho et al. Thin Hybrid Metamaterial Slab With Negative and Zero Permeability for High Efficiency and Low Electromagnetic Field in Wireless Power Transfer Systems. *IEEE Trans. Electromagn. Compat.*, vol. 60, no. 4, pp. 1001-1009, Aug. 2018. <https://doi.org/10.1109/TEM.2017.2751595>.
- [R12] Y. Chen, X. Zhao, S. Niu, W. Fu and H. Lin. A Transmitter-Embedded Metasurface-Based Wireless Power Transfer System for Extended-Distance Applications. *IEEE Trans. Power. Electron.*, vol. 39, no. 1, pp. 1762-1772, Jan. 2024, <https://doi.org/10.1109/TPEL.2023.3320743>.
- [R13] R. Das, A. Basir and H. Yoo, "A Metamaterial-Coupled Wireless Power Transfer System Based on Cubic High-Dielectric Resonators," *IEEE Trans. Ind. Electron.*, vol. 66, no. 9, pp. 7397-7406, Sept. 2019. <https://doi.org/10.1109/TIE.2018.2879310>.
- [R14] L. Li, H. Liu, H. Zhang and W. Xue, "Efficient Wireless Power Transfer System Integrating With Metasurface for Biological Applications," *IEEE Trans. Ind. Electron.*, vol. 65, no. 4, pp. 3230-3239, April 2018. <https://doi.org/10.1109/TIE.2017.2756580>.

- [R15] W. Lee and Y. -K. Yoon, "Tunable Metamaterial Slab for Efficiency Improvement in Misaligned Wireless Power Transfer," *IEEE Microw. Wirel. Compon. Lett.*, vol. 30, no. 9, pp. 912-915, Sept. 2020, <https://doi.org/10.1109/LMWC.2020.3015680>.
- [R16] M. Aboualalaa and R. K. Pokharel, "Reliable Multiple Cascaded Resonators WPT System Using Stacked Split-Ring Metamaterial Passive Relays," *IEEE Trans. Instrum. Meas.*, vol. 72, pp. 1-10, 2023.<https://doi.org/10.1109/TIM.2023.3324672>.
- [R17] K. Lee and S. H. Chae, "Power Transfer Efficiency Analysis of Intermediate-Resonator for Wireless Power Transfer," *IEEE Trans. Power. Electron.*, vol. 33, no. 3, pp. 2484-2493, March 2018.<https://doi.org/10.1109/TPEL.2023.3320743>.
- [R18] X. Fan, F. Tang, B. Su and X. Zhang, "Design of Spiral Resonator Based on Fractal Metamaterials and Its Improvement for MCR-WPT Performance," *IEEE Trans. Magn.*, vol. 58, no. 8, pp. 1-9, Aug. 2022.<https://doi.org/10.1109/TMAG.2022.3186089>.
- [R19] M. -L. Kung and K. -H. Lin, "Dual-Band Coil Module With Repeaters for Diverse Wireless Power Transfer Applications," *IEEE Trans. Microw. Theory Tech.*, vol. 66, no. 1, pp. 332-345, Jan. 2018.<https://doi.org/10.1109/TMTT.2017.2711010>.

Changes Made: Table R2 is supplemented in the revised manuscript as Table 4. The advantages of the proposed NEMR structure have been supplemented in the last paragraph on page 3 of the revised manuscript.

Thank you so much for your time and comments!

Reviewer 2:

The manuscript is based on a magnetic circuit model of a metamaterial embedded with transmitter and receiver coils in a Wireless Power Transfer system for calculating the reluctance and mutual inductance of the metamaterial. Unfortunately, the magnetic circuit model, from my point of view, is not well-described and explained. Furthermore, the experimental verification is not clear.

Response of the overall comment.

The authors would like to thank Reviewer 2 for the constructive comments and suggestions and would like to provide the following explanations for the comments. The authors have carefully revised this manuscript according to the comments. The changes in the revised manuscript are highlighted in red colour.

Comment-1. The overall presentation of the work confuses the reader. The authors mention a stacked version of the metamaterial, in the abstract, but finally they do not use it for experimental verification.

Response-1. Thank you for pointing it out. The stacked version of the NEMR structure is used as the magnetic core of the transformer, to verify that the equivalent magnetic reluctance of the structure can be negative and be influenced by its volume. The experiments are given in Part A of Section IV of the manuscript. To avoid misunderstanding, the description of the stacked NEMR structure experiments is reorganized, which clearly illustrates the impact of magnetic reluctance on the voltage proportion of the transformer with NEMR structures.

Excepting the stacked NEMR structure version, the single layer NEMR structure experiments are also conducted. The single layer NEMR structure is installed in the transmitter coil and receiver coil of the WPT system, to verify that the structure can increase the mutual inductance and efficiency of the weak coupling WPT system.

Changes Made: The description of the stacked NEMR structure in ABSTRACT is added based on the comment.

Comment-2. The following sentence is not clear: "However, considering that the hysteresis loss of generalized iron-oxide ferrite will boost if the operating frequency of the WPT system is higher than the rated frequency" I think some verbs of conclusions are missing.

Response-2: Thank you for pointing it out. The conclusion of this sentence is supplemented and the sentence is revised as 'However, the hysteresis loss of the iron-oxide ferrite will boost when the system frequency exceeds the working frequency range of ferrite, leading to a decrease in the efficiency of the WPT system "

Changes Made: The mentioned sentence in the INTRODUCTION has been corrected in the revised manuscript.

Comment-3. The following sentences are not very clear about what they refer to. Please, revise accordingly. "However, those three types of solutions both have some inherent drawbacks. First, the hysteresis loss of the magnetic ferrite core will boost with the increase in operating frequency." Probably, some conclusions are missing.

Response-3: Thank you for your suggestion, the mentioned sentence is reorganized and the conclusion of this paragraph is supplemented, given as follows.

However, increasing operating frequency or employing unconventional ferrite material both have some inherent drawbacks. As for the former, the hysteresis loss of the ferrite would boost when the system frequency exceeds the working frequency range of the ferrite, making it inapplicable to high-frequency weak-coupling WPT systems. For employing unconventional materials, the effectiveness of efficiency enhancement is limited. Besides, due to the positive permeability of ferrite materials, regardless of optimization and design, the corresponding magnetic reluctance always remains positive. Consequently, in terms of magnetic reluctance reduction for WPT systems, the ferrite material is inherently weaker than metamaterial with negative permeability. For the reasons given above, the metamaterial is considered a potentially ideal solution for high-frequency weak-coupling WPT systems.

Changes Made: The third paragraph of the INTRODUCTION in the manuscript has been revised.

Comment-4. Page 2: "increases" instead "increase"?

Response-3: Thank you for pointing it out. The sentence is corrected as 'Ref. [19] employed the designed $NdxFe_{1-x}Ny$ material as the magnetic core of a 13.56 MHz system, which **increases** the inductance from 0.69 to 1.15 μH .

Changes Made: The mistake on page 2 has been corrected.

Comment-5. This sentence "This issue results that the high operating frequency and magnetic ferrite core can not be applied in weak coupling WPT systems simultaneously." I think is redundant.

Response-5: Thank you for the suggestion, the sentence is written as 'As for the former, the hysteresis loss of the ferrite will boost when the system frequency exceeds the working frequency range of ferrite, making it inapplicable to high frequency weakly coupled WPT systems.'

Changes Made: The sentence in the third paragraph of INTRODUCTION on page 2 has been revised.

Comment-6. The equivalent magnetic circuit of the system is not clear. The authors define five reluctances parameters R_{p0} , R_{p1} , R_{p2} , R_{p3} and R_{p4} , which they describe as the magnetic reluctance of the branches of the transmitter coil. What the branches of the transmitter coil are? (SubQ1) Why the reluctances parameters are in a number of five? Please, revise and explain accordingly. (SubQ2) A same clarification is required for parameters R_{m0} , R_{m1} and R_{m2} and R_{core} (SubQ3).

Response-6: To better illustrate this issue, **Comment-6** is divided into three separate questions.

SubQ1. The magnetic reluctance R_{p0} , R_{p1} , R_{p2} , R_{p3} and R_{p4} (R_{p1} , R_{p2} , R_{p3} and R_{p4} are redefined as R_{l1} , R_{l2} , R_{l3} and R_{l4} , respectively while R_{p0} is removed) are not the branches of the transmitter coil. Those magnetic reluctances form the magnetic circuit of the leakage inductance.

To better analyze the magnetic circuit of the system, the description and magnetic circuit in the manuscript have been revised from Fig. R1(a) to Fig. R1(b).

Fig. R1. The equivalent magnetic circuit of the proposed design. (a) Original magnetic circuit analysis. (b) Revised magnetic circuit analysis

The total flux ϕ_p generated by the transmitter coil consists of the mutual flux ϕ_m and the leakage flux ϕ_l . The relationships between those fluxes based on Fig. R1(b) are given as follows.

$$\phi_p = \phi_m + \phi_l = \underbrace{\phi_{m1} + \phi_{m2}}_{\text{Mutual flux}} + \underbrace{\phi_{l1} + \phi_{l2}}_{\text{leakage flux}} \quad (R1)$$

where ϕ_{m1}, ϕ_{m2} are the flux in two branches of the magnetic path of mutual flux, respectively; ϕ_{l1}, ϕ_{l2} are that of leakage flux.

The magnetic circuit of leakage flux ϕ_l contains the magnetic reluctance R_{l1}, R_{l2}, R_{l3} and R_{l4} . R_{l1} and R_{l2} represent the equivalent magnetic reluctances of the area between the transmitter coil and the receiver coil, respectively. R_{l3} and R_{l4} represent the equivalent magnetic reluctances of the remaining area outside the abovementioned area. Equivalently, R_m, R_{m1} , and R_{m2} are the magnetic reluctance corresponding to the mutual inductance M , which represents that outside the receiver coil. R_{core} is the equivalent magnetic reluctance of the core determined by its material.

SubQ2. The five reluctance parameters are revised as four parameters, which are shown in Fig. R1 (b). The clarification of magnetic reluctances is based on the corresponding flux.

i.e. For the leakage flux ϕ_l , the corresponding magnetic circuit is divided into two branches. R_{l1} and R_{l2} represent the equivalent magnetic reluctances of the area between the transmitter coil and the receiver coil, respectively. R_{l3} and R_{l4} represent the equivalent magnetic reluctances of the remaining space outside the abovementioned area.

SubQ3. Similar to the clarification of R_{l1}, R_{l2}, R_{l3} and R_{l4} which correspond to the leakage fluxes, R_m, R_{m1} , and R_{m2} are the magnetic reluctance corresponding to the mutual inductance M . These terms symbolize the magnetic reluctance of two branches of the magnetic circuit external to the receiver coil. R_{core} is the equivalent magnetic reluctance of the core material, which is determined by its electromagnetic property.

Changes Made: i) Fig. 2 of the manuscript is revised as Fig. R1. (b). ii) The clarification of the magnetic reluctance is redefined and rewritten. iii) The equivalent magnetic circuit analysis is revised.

Comment-7. In table II, the authors define a compensation capacitor, which is the first time they introduce in the manuscript. Please, could you explain in which part of their system they are using a compensation capacitor? (SubQ1) Probably the equivalent electrical circuit of the system needs to be presented before (SubQ2).

Response-7: To better illustrate this issue, **Comment-7** is divided into two separate questions.

SubQ1. In the wireless power transfer system, the transmitter coil and the receiver coil require the resonant network, including LC, LLC, and CLLC resonant network [R1]. To achieve a high transfer efficiency, the transmitter coil and receiver coil must be connected to a compensation capacitor and/or compensation inductor [R2]. The most commonly used composition resonant network is concluded in Fig. R2. As shown in Fig. R2, based on the compensation topology in the transmitter coil (primary side) and receiver coil (secondary side), the compensation network is mainly divided as the series-series topology (SS) in Fig. R2(a), series-parallel topology (SP) in Fig. R2(b), parallel-series topology (PS) in Fig. R2(c), and parallel-parallel topology (PP) in Fig. R2(d).

Fig. R2. The most widely used topology in wireless power transfer systems. (a) Series-series topology. (b) Series-parallel topology. (c) Parallel-series topology. (d) Parallel-parallel topology.

where R_p , R_s is the internal resistance of the transmitter coil and receiver coil, respectively; C_p , C_s is the compensation capacitor of the transmitter coil and receiver coil, respectively; R_L is the load of the receiver coil; L_p , L_s are the self-inductance while M is the mutual inductance of those two coils; L_x is the compensation inductor and v_{in} is the input voltage.

The compensation network and output property of those four topologies are summarized as follows.

Table. R1. The operating condition and output properties of compensation topologies.

Compensation topology	Operating condition	Output properties	Output current/voltage
Series-Series	$\omega = \frac{1}{\sqrt{L_p C_p}} = \frac{1}{\sqrt{L_s C_s}}$	Constant current output	$i_o = \frac{v_{in}}{\omega M}$
Series-Parallel	$\omega = \frac{1}{\sqrt{L_p - \frac{M^2}{L_s} C_p}} = \frac{1}{\sqrt{L_s C_s}}$	Constant voltage output	$v_o = \frac{L_s v_{in}}{M}$
Parallel-Series	$\omega = \frac{1}{\sqrt{L_p C_p}} = \frac{1}{\sqrt{L_s C_s}}, L_x = L_p$	Constant voltage output	$v_o = \frac{M v_{in}}{L_x}$
Parallel-Parallel	$\omega = \frac{1}{\sqrt{L_p - \frac{M^2}{L_s} C_p}} = \frac{1}{\sqrt{L_s C_s}}, L_x = L_p - \frac{M^2}{L_s}$	Constant current output	$i_o = \frac{M v_{in}}{\omega L_x L_s}$

where ω is the resonance frequency of the WPT system.

Based on Table R1, in this paper, the series-series topology (SS) is selected as the compensation topology of the wireless power transfer system, considering the output of SS topology is only determined by both the mutual inductance M and operating frequency ω .

SubQ2. Thank you for pointing it out. To avoid the description misleading, Table 2 is divided into two separate Tables (Table 2 and Table 3 in Section 4 of the revised manuscript), to present the geometrical parameters and the electric parameters of the proposed NEMR structure-based WPT system.

[R1] W. Zhang and C. C. Mi, "Compensation Topologies of High-Power Wireless Power Transfer Systems," *IEEE Trans. Veh. Technol.*, vol. 65, no. 6, pp. 4768-4778, June 2016, doi: 10.1109/TVT.2015.2454292. doi: 10.1109/TVT.2015.2454292.

[R2] K. N. Mude and K. Aditya, "Comprehensive review and analysis of two-element resonant compensation topologies for wireless inductive power transfer systems," *Chinese Journal of Electrical Engineering*, vol. 5, no. 2, pp. 14-31, June 2019. doi: 10.23919/CJEE.2019.000008.

Changes Made: Table 2 has been divided into two separate Tables in the revised manuscript, named GEOMETRIC PARAMETERS OF THE COILS AND NEMR STRUCTURE (Table 2) and ELECTRIC PARAMETERS OF THE COILS AND NEMR STRUCTURE (Table 3).

Comment-8. Equation (4): the value assumed from the reluctance in air I think is not correct. Instead of μ , you probably mean " μ_0 "?

Response-8: Thank you for pointing it out. The permeability of all materials consists of the relative permeability μ_r and the vacuum permeability μ_0 , defined as follows

$$\mu = \mu_r \mu_0 \tag{R2}$$

For the air, the relative permeability μ_r is 1. Hence, the permeability μ of the air equals μ_0 .

To prevent the misunderstanding, Eq. (4) (Eq.(5) in the revised manuscript) has been corrected.

Changes Made: The definition of the permeability of the air in Eq.(4) (Eq. (5) of the revised manuscript) has been corrected as follows.

$$R_{core} = \frac{l}{\mu_r \mu_0 A} \approx \begin{cases} \frac{l}{\mu_0 A}, air \\ R_{fer}, ferrite material \\ R_{NEMR}, NEMR structure \end{cases} \tag{R3}$$

where μ_0 and μ_r are the vacuum and relative permeability of the core material, respectively. R_{fer} and R_{NEMR} are the magnetic reluctance of ferrite material and NEMR structure. Considering the relative permeability μ_r of air is 1, the magnetic reluctance of air is expressed as $\frac{l}{\mu_0 A}$.

Comment-9. Page 7: "Rfer" instead of "Rref"?

Response-9: Thank you for pointing it out. R_{fer} refers to the magnetic reluctance of the ferrite material. The typo R_{ref} on page 7 has been corrected.

Changes Made: The mistakes on page 7 of the manuscript have been revised as follows.

As shown in Eqs. (5) and (6), the mutual inductance of the two-coil system is inversely proportional to R_{Core} . Hence, if the R_{NEMR} is lower than R_{fer} , the effect of NEMR structure on mutual inductance enhancement is better than that of ferrite material theoretically.

Comment-10. Equation (6): Definition for $P_{in} = U_s I_p$ is not precise from electrical circuit theory point of view. Please revise considering that the equation are referred to an AC circuit.

Response-10: Thank you so much for pointing it out, the quality factor $\cos(\theta)$ of the primary side (transmitter coil) is ignored. To address this issue, Eq. (6) (Eq.(7) of the revised manuscript) is revised as Eq. (R4) based on the AC circuit theory.

The input power P_{in} and the output power P_{out} of the WPT system under the rated operating condition are as follows.

$$\begin{cases} P_{in} = U_s I_p \cos(\theta) = I_p^2 \left(R_p + j\omega L_p + \frac{1}{j\omega C_p} + \frac{\omega^2 M^2}{R_s + R_L + j\omega L_s + \frac{1}{j\omega C_s}} \right) \\ P_{out} = I_s^2 R_L = \frac{\omega^2 M^2}{\left(R_s + R_L + j\omega L_s + \frac{1}{j\omega C_s} \right)^2} I_p^2 R_L \end{cases} \quad (R4)$$

where θ is the phase difference between the primary voltage and current

Under the rated operating condition, the frequency ω equals the resonant frequency ω_o of the system, which should meet the requirements of $\omega_o = \frac{1}{\sqrt{L_p C_p}} = \frac{1}{\sqrt{L_c C_c}}$ and makes the summation of reactive component $(j\omega L_p + \frac{1}{j\omega C_p}$ and $j\omega L_s + \frac{1}{j\omega C_s})$ zero. Under this condition, the phase difference θ is near zero, making $\cos(\theta)$ equals one.

Changes Made: Eq. (6) of the original manuscript has been revised as Eq. (R4) and renumbered as Eq. (7).

Comment-11. Equation (8): for defining the efficiency, the authors cite a manuscript, [22]. Actually, in [22] it seems that there is not a definition of the efficiency. Please, revise accordingly.

Response-11: Thank you for pointing it out, the misquotation of Ref. [22] has been revised. The equation (8) is from ref. [R2] (Ref. [35] of the revised manuscript).

[R3] J. H. Kim et al., "Development of 1-MW Inductive Power Transfer System for a High-Speed Train," *IEEE Trans. Ind. Electron.*, vol. 62, no. 10, pp. 6242-6250, Oct. 2015. <https://doi.org/10.1109/TIE.2015.2417122>.

Changes Made: The mistakes on page 3 of the manuscript have been corrected and Ref. [22] in the revised manuscript is revised as Ref.[35].

Comment-12. The authors selected the operational frequency equal to 6.78MHz. Please, could you explain why they choose this frequency?

Response-12: Thank you for your comment. the operating frequency is selected as 6.78 MHz based on the Rezence standard [R4] developed by Alliance for Wireless Power (A4WP).

[R4] Wikipedia contributors. Rezence (wireless charging standard). Wikipedia. [https://en.wikipedia.org/wiki/Rezence_\(wireless_charging_standard\)](https://en.wikipedia.org/wiki/Rezence_(wireless_charging_standard))

Comment-13. Equation (9) is completely uncorrect from conceptual point of view. The authors mixed a trattation in phasorial regime and temporale regime. Please revise accordingly.

Response-13: Thank you for pointing it out, the number of turns N of the NEMR structure is incorrectly reused in both Eq. (R5) and (R6) [Eqs. (9) and (10) of the manuscript]. The description and the equations have been corrected as follows.

As for the equivalent circuit of the NEMR structure, the corresponding induced voltage equals the product of current and total impedance, given in Eq. (R5) [Eq. (9) in the manuscript].

$$I \left(R + \frac{1}{j\omega C} + j\omega L \right) = U_{ind} \quad (R5)$$

where R , C , and L , are the circuit parameters of the NEMR structure. U_{ind} is the induced voltage of the NEMR structure caused by the flux variation $d\phi$, which can also be defined as Eq. (R6) [Eq. (10) in the manuscript]. The flux variation $d\phi$ is determined by the flux generated by the transmitter, receiver and/or another NEMR structure.

$$U_{ind} = N \frac{d\phi}{dt} = \frac{dB}{dt} \sum_{k=1}^N S_k \quad (R6)$$

where N is the number of turns and S_k is the corresponding equivalent area of each turn of the NEMR structure.

Changes Made: Eqs. (9), (10) and corresponding descriptions have been carefully revised.

Comment-14. It is not clear the design of the system. What the value for parameter n is?. Please clarify this aspect.

Response-14: Thank you for pointing the typo out. n is the number of turns of the NEMR structure, which should be N instead of n .

Changes Made: The typo about the number of turns of the NEMR N has been revised.

Comment-15. Equation (14): What the parameter A is in the equation? Please, explain.

Response-15: Thank you for the comment. A is the cross-sectional area of the structure.

Changes Made: The description of Eq. (14) is revised as 'where A is the cross-sectional area of the circuit in square metres. ω is the operating frequency of the WPT system while ω_0 is the resonance frequency of the NEMR structure with compensation capacitor C .

Comment-16. Electromagnetic analysis section: please give more information about the numerical analysis you performed. Which software used, the EM meshgrid, and so the number of unknowns, the hardware performance they used for simulations.

Response-16: Thank you for your comment. The information about the numerical analysis is given as follows.

The simulation is conducted via the Ansys HFSS. The input power of the transmitter coil is set as 1 W and the load in the receiver coil is selected as 50 ohm. Considering the load of those four systems and the input power is constant, the higher current in the receiver coil indicates a higher receiver power and efficiency. The magnetic induction B is directly proportional to the current I in the conductor (receiver) based on Ampère's circuital law, which equals the product of magnetic field strength H and permeability μ .

The mesh length is selected as 15 mm and the mesh distribution is indicated in Fig. R3. The total number of elements of the model is 272641 and the solution domain contains 249278 elements

Total number of elements: 272641									
	Num Tets	Min edge length	Max edge length	RMS edge length	Min tet vol	Max tet vol	Mean tet vol	Std Devn (vol)	
Cylinder1	249278	0.107471	49.3878	10.3532	1.48006e-06	6686.69	104.969	336.24	
Octagon_Rx	3651	0.71149	17.9252	4.8812	0.00329918	0.746882	0.175431	0.0843299	
Octagon_Tx	3665	0.707107	23.6992	4.86006	0.00179195	0.987467	0.17476	0.0825199	
Rx	8069	0.76283	9.59778	3.48805	9.9517e-06	1.63367	0.491118	0.209779	
Tx	7978	0.577447	9.83316	3.52001	4.78805e-06	1.69168	0.49672	0.207616	

Fig. R3. The mesh distribution of the model of the NEMR structure-based WPT system

Changes Made: The description of the simulation has been supplemented in Section III of the revised manuscript.

Comment-17. Fig.6: explaining the results of figure 6, the authors say that "As shown in Fig. 6, the WPT system with the dual coil-embedded NEMR structure has the best performance,". I actually do not understand what is the difference between the field maps presented in Fig. 6. They look very similar each others. (SubQ1) Please, clarify this aspect and I suggest to use a colormap scale which highlights the differences, if there are. (SubQ2)What about the simulated efficiency of the system?(SubQ3) Is it comparable with the results of experimental analysis?(SubQ4)

Response-17: Thank you for your comment. To better illustrate this issue, **Comment-17** is divided into four separate questions.

SubQ1. Thank you for your comment. To better illustrate the difference between systems with and without the NEMR structure. Fig. 6 of the manuscript is revised as Fig. R4 and the corresponding description is revised as follows.

The simulation is conducted via the Ansys HFSS. The input power of the transmitter coil is set as 1 W and the load in the receiver coil is selected as 50 ohm. Considering the load of those four systems [i) system without NEMR structure, ii) system with NEMR structure in the transmitter coil, iii) system with NEMR structure in the receiver coil, iv) system with NEMR structure in both the transmitter and receiver coil], and the input power is constant, the higher current in the receiver coil indicates a higher receiver power and efficiency.

The magnetic field intensity H is directly proportional to the current I in a conductor (receiver) based on Ampère's circuital law, described as follows.

$$\int_C H dl = I \quad (R7)$$

Considering that the input power of the transmitter coil is constant, by comparing the magnetic field strength H around the receiver coil, the current I in the receiver coil can be obtained and the efficiency comparison between WPT systems can be found roughly.

As shown in Fig. R4, the WPT systems with the dual coil-embedded NEMR structure has the best performance (highest magnetic field strength H around receiver), followed by the system with transmitter-embedded, receiver-embedded NEMR structure, as well as the system without the NEMR structure.

The NEMR structures increase the magnetic field intensity H around the receiver coil to a different extent, which is directly connected to the power transfer efficiency of the WPT system.

Fig. R4. The magnetic field strength distribution. (a) Without NEMR structure. (b) With the transmitter-embedded NEMR structure. (c) With the receiver-embedded NEMR structure. (d) With dual coil-embedded NEMR structure.

SubQ2. Thank you for your advice, the colour map scale is supplemented in the first column of Fig. R4 to highlight the difference between the magnetic field strength H around the receiver.

SubQ3. The simulation efficiency is given in Figs. R5 and R6. As shown in Fig. R5, the efficiency of the WPT system is increased from 33.28% to 43.54% under the transfer distance of 150 mm. The NEMR structure can effectively increase the efficiency of the weak coupling WPT system. Even though there are some differences between the simulation and experimental results, the effectiveness of the NEMR structure in efficiency enhancement is evident.

This could be because HFSS software is usually used for high frequency (GHz) simulation, and the computational accuracy in the MHz region is not very high. Hence, the Ansys HFSS simulation is used to study the magnetic field distribution of the WPT system with/without the NEMR structures.

Fig. R5. The efficiency of the WPT system under the transfer distance of 150 mm. (a) The system without NEMR structure. (b) The system with dual-NEMR structures.

Fig. R6 The efficiency of the WPT system with the different transfer distance.

In practical application, the efficiency is impacted by the load condition and power level. Considering the conditions mentioned above, the experimental efficiency is selected and indicated in this manuscript.

SubQ4. Thank you for your comment. The simulation results indicate the same trend as the experimental results. Considering the computational accuracy of the software and platform of the simulation, the efficiency verified via experimental results is given in the manuscript.

Changes Made: i) The simulation condition has been supplemented in Section III of the manuscript. ii) Fig. 6 of the manuscript is revised as Fig. R4 and the corresponding description has been revised.

Comment-18. Fig.8: "source" instead of "sourse"

Response-18: Thank you for pointing it out, the typo in Fig. 8 of the manuscript has been revised.

Changes Made: Fig. 8 of the manuscript has been corrected.

Comment-19. Equation (16): the sentence "equal the corresponding magnetic reluctance excepting that of core material in Eq. (3)" has not sense.

Response-19: Thank you for pointing it out, to avoid misunderstanding, the description for the transformer with NEMR structure-based core is revised and given as follows.

The experiment platform and the equivalent magnetic circuit of the designed transformer are shown in Figs. R7(a), (b), and (c).

Fig. R7. The transformer-based verification system for negative magnetic reluctance verification. (a) NEMR structure-based transformer. (b) Equivalent circuit. (c) Verification system.

The voltage proportion between the primary coil and the secondary coil is determined by the self-inductance, mutual inductance, and current in coils, given in Eq. (R8).

$$\begin{cases} U_s = j\omega L_p i_p + j\omega M i_s + i_p R_p \\ U_o = j\omega M i_p + j\omega L_s i_s + i_s R_s \end{cases} \quad (R8)$$

where $U_s, U_o, L_p, L_s, i_p, i_s R_p, R_s$ are the voltage, self-inductance, current, and resistance of the primary side and secondary side, respectively; M is the mutual inductance of two coils.

For the no-load operating condition, the secondary current i_s is considered as zero. The relationship between the input voltage U_s and output voltage U_o can be found as follows.

$$\frac{U_o}{U_s} = \frac{j\omega M i_p + j\omega L_s i_s + i_s R_s}{j\omega L_p i_p + j\omega M i_s + i_p R_p} = \frac{j\omega M}{j\omega L_p + R_p} \quad (R9)$$

In this transformer, the total magnetic reluctance R_{mt} of the transformer, including the self-inductance L_p and mutual inductance M , consists of the magnetic reluctance R_{NEMRS} of the stacked NEMR structure and that of the air R_{Air} . Based on the definition of inductance, considering that the number of turns of the primary coil and secondary coil are equal, Eq. (R9) can be written as follows.

$$\frac{U_o}{U_s} \approx \frac{j\omega N^2}{j\omega N^2 + R_p R_{mt}} \quad (R10)$$

As given in Eq. (R10), when the total magnetic reluctance R_{mt} is negative, the voltage proportion will increase and be larger than one.

Changes Made: The description of the experiments of the NEMR structure-based transformer is revised and given in part A of Section IV of the revised manuscript.

Comment-20. The experimental analysis is a little bit confusing the reader. The authors examine a stacked version of the NEMR structure, mentioned only in the abstract. Please, clarify this aspect. Did the authors perform electromagnetic simulation of the stacked version of the NEMR structure? (SubQ1) Could you explain why do you use a stacked version of NEMR? (SubQ2) What is the distance between each layer? Please, give more details. (SubQ3)

Response-20: Thank you for your comment. To better illustrate this issue, **Comment-20** is divided into three separate questions.

SubQ1. The electromagnetic simulation of the stacked version of the NEMR structure is not conducted. The stacked NEMR structure consists of 15 single-layer NEMR structures. The single layer NEMR structure is the same as the NEMR structure used in the WPT system, which is indicated in Fig. R8.

Fig. R8. The transformer with NEMR structure core and single layer NEMR structure in the WPT system. (a) The core of the transformer. (b) the single layer NEMR structure & the transmitter/receiver coil in the WPT system with NEMR structure

SubQ2. The stacked version of the NEMR structure is used to verify that NEMR can indicate a negative magnetic reluctance and that the magnetic reluctance is influenced by the volume of the NEMR structure.

As indicated in Eq. (R10), the voltage proportion between the input voltage and the output voltage will increase and be larger than 1 when the total magnetic reluctance R_{mt} .

To verify that the magnetic reluctance of the stacked NEMR structure is influenced by the volume, the experiments of the transformer with the different layers of stacked NEMR structure core are conducted. The experiment results are given in Fig. R9. Experimental results indicate that the proportion will increase with the increase of the number of layers/volume of the stacked NEMR structure. Compared to the stacked NEMR structure with 15 layers, the peak voltage proportion of the stacked NEMR structure with 13 and 11 layers will decrease from 8.31 to 7.62 and 7.15, respectively. Hence, the magnetic reluctance of the stacked NEMR structure is influenced by the volume.

Fig. R9. The voltage proportion versus the operating frequency.

SubQ3. The distance between the NEMR structure is considered as zero, considering that each slab is close to each other. The schematic diagram of the stacked NEMR structure is given in Fig. R8(a), and the NEMR structure core is

Fig. R10. The schematic diagram of the stacked NEMR structures in transformer. (a) The NEMR structures-based transformer. (b) The core of the transformer (stacked NEMR structure) in front view. (c) The core of the transformer (stacked NEMR structure) in 45 degree top view. (d) Single layer NEMR structure.

given in Fig. R10. The stacked NEMR structure consists of 15 identical single-layer NEMR structures, where the single-layer NEMR structure is used in the WPT systems.

Changes Made: The description of the transformer with the stacked NEMR structures has been carefully revised in part A of Section IV of the revised manuscript.

Comment-21. The transmission gain verification is performed with a stacked NEMR structure or a single NEMR? (SubQ1) From figures, It looks like that the verification has been performed with a not stacked version of NEMR. Why? Please, clarify this aspect.(SubQ2)

Response-21: Thank you for your comment. To better illustrate this issue, **Comment-21** is divided into two separate questions.

SubQ1. The transmission gain verification is performed with a single-layer NEMR structure. The transmission gain is an effective method to measure the efficiency of the WPT system [R3]-[R5].

SubQ2. The staked NEMR structure forms the core of the transformer, to verify that the NEMR can indicate a negative magnetic reluctance and that the magnetic reluctance of the stacked NEMR structure is influenced by the volume. This is observed by the proportion between the input voltage and output voltage of the transformer under different frequencies, which is given in Fig. R9.

The single-layer NEMR structure is used to increase the mutual inductance and the efficiency of the weak coupling WPT systems. This is verified by the transmission gain, open circuit experiment and power transfer efficiency experiments.

To sum up, the stacked NEMR structure is used to verify the properties of the NEMR structure while the single layer NEMR structure is used to verify its effectiveness in WPT applications.

[R3] C. Lu, X. Huang, C. Rong, X. Tao, Y. Zeng and M. Liu, "A Dual-Band Negative Permeability and Near-Zero Permeability Metamaterials for Wireless Power Transfer System," *IEEE Trans. Ind. Electron.*, vol. 68, no. 8, pp. 7072-7082, Aug. 2021.doi: 10.1109/TIE.2020.3009608.

[R4] M. Aboualalaa and R. K. Pokharel, "Reliable Multiple Cascaded Resonators WPT System Using Stacked Split-Ring Metamaterial Passive Relays," *IEEE Trans. Instrum. Meas.*, vol. 72, pp. 1-10, 2023.doi: 10.1109/TIM.2023.3324672.

[R5] R. Das, A. Basir and H. Yoo, "A Metamaterial-Coupled Wireless Power Transfer System Based on Cubic High-Dielectric Resonators," *IEEE Trans. Ind. Electron.*, vol. 66, no. 9, pp. 7397-7406, Sept. 2019.doi: 10.1109/TIM.2023.3324672.

Comment-22. Conclusions section: the value for mutual inductance improvement is not readable.

Response-22: Thank you for pointing it out, we have carefully revised the conclusion. The corresponding sentence is revised as 'the mutual inductance enhancement property of the NEMR structure in weak coupling WPT system is verified, which can increase the mutual inductance by more than 154.2%.'

Changes Made: The conclusion has been carefully revised based on the reviewer's comment.

Thank you so much for your comments on this manuscript! Your comments are invaluable in improving the quality of this manuscript.

REVIEWER COMMENTS

Reviewer #1 (Remarks to the Author):

All of my concerns were addressed, and I am satisfied with the modifications.

Reviewer #2 (Remarks to the Author):

I thank the authors for their impressive work on performing the revisions of the manuscript. Now everything is more clear and the manuscript has much improved.

However, I have to submit you some issues:

-Response 6 to my comment 6: I really appreciated the explanation, but I still don't understand why the authors decided to change R_{p1} (in the original version) with a parallel version of R_{l1} ?

Please clarify.

-Response 9 to my comment 9: You are still using the same quantity (symbol) U_{ind} , to describe two different behaviour. In (9) U_{ind} is a phasor, in (10) U_{ind} is a quantity varyng with time. Please revise accordingly.

-Reponse 16 to my comment 16: a better clarification needs to be performed to pass from (R9) to (R10)

Dear Reviewers of Nature Communication,

Original ID NCOMMS-23-52178A entitled 'Modelling of Negative Equivalent Magnetic Reluctance Structure and Its Application in Weak-Coupling WPT Systems'.

Thank you for your comments on our manuscript. We have revised our manuscript according to your comments and suggestions. All the specific suggestions and remarks provided by the reviewers have been incorporated in the revised version of the manuscript. The changes in the revised manuscript are highlighted in **red color** in this new version of the manuscript.

In summary, the following changes are applied to the submitted manuscript:

- 1) Based on the comments, the description of the magnetic circuit of Fig. 2 in the manuscript has been revised.
- 2) Eqs. (9) and (10) have been corrected.
- 3) A detailed clarification of Eqs. (16)-(18) in the second round revised manuscript [which are the Eqs. (R9) and (R10) in the first round revision] has been supplemented.
- 4) The manuscript has been carefully revised.

Response to Reviewers:

Reviewer 1:

The authors would like to thank Reviewer 1 for the constructive comments and suggestions.

Comment-1. All of my concerns were addressed, and I am satisfied with the modifications.

Response-1: Thank you for your time to review our manuscript.

Thank you so much for your time and comments!

Reviewer 2:

I thank the authors for their impressive work on performing the revisions of the manuscript. Now everything is more clear and the manuscript has much improved. However, I have to submit you some issues.

Response of the overall comment.

The authors would like to thank Reviewer 2 for the constructive comments and suggestions and would like to provide the following explanations for the comments. The authors have carefully revised this manuscript according to the comments. The changes in the second round revised manuscript are highlighted in red colour.

Comment-1. Response 6 to my comment 6: I really appreciated the explanation, but I still don't understand why the authors decided to change R_{p1} (in the original version) with a parallel version of R_{l1} ? Please clarify.

Response-1. Thank you for pointing it out. The figure in Response 6 is given as Fig. SR1.

Fig. SR1. The equivalent magnetic circuit of the proposed design. (a) Original magnetic circuit analysis. (b) Revised magnetic circuit analysis

The change from R_{p1} to R_{l1} is with the consideration of the **distribution of the magnetic field** and the **specifications for total flux, leakage flux and mutual flux**. In Fig. SR1(b), for the magnetic flux, the subscript l refers to the leakage flux, the subscript m refers to the mutual flux and the subscript p refers to the total flux. The specification of magnetic reluctance is the same as that of flux. As for the **distribution of the magnetic field**, a conductor with the current will generate a closed magnetic field consisting of numbers of the magnetic flux lines, which is indicated in Fig. SR2.

Fig. SR2. The schematic diagram of the magnetic flux/path distribution of the proposed system.

The relationships of flux in Fig. SR2 are concluded as follows.

$$\begin{cases} \phi_p = \phi_{p1} + \phi_{p2} \\ \phi_m = \phi_{m1} + \phi_{m2} \\ \phi_{p1} = \phi_{m1} + \phi_{l1} \\ \phi_{p2} = \phi_{m2} + \phi_{l2} \end{cases} \quad (SR1)$$

where ϕ_p is the total flux generated by the transmitter coil; ϕ_m is the mutual flux while $\phi_{l1} + \phi_{l2}$ is the leakage flux. $\phi_{p1}, \phi_{p2}, \phi_{l1}, \phi_{l2}, \phi_{m1}$, and ϕ_{m2} are the corresponding flux in branches of the magnetic path.

The magnetic field distribution consists of many closed magnetic flux lines, as shown in Fig. RS2. Each magnetic flux line corresponds to a magnetic circuit/path. **R_{l1} and R_{l2} are used to represent that the magnetic reluctance in the branches is composed of the parallel combination of the magnetic reluctances.**

In the former description of magnetic circuits with R_{p1} and R_{p2} in Fig. SR1(a), the parallel combination of the magnetic reluctances was **not** presented.

Changes Made: The description of the magnetic circuits in Fig. 2 of the second round revised manuscript has been revised.

Comment-2. Response 9 to my comment 9: You are still using the same quantity (symbol) U_{ind} , to describe two different behaviour. In (9) U_{ind} is a phasor, in (10) U_{ind} is a quantity varying with time. Please revise accordingly.

Response-2: Thank you for pointing it out. The symbols of the induced voltages of the negative equivalent magnetic reluctance have been revised, given as follows.

As for the equivalent circuit of the NEMR structure, the corresponding induced voltage equals the product of current and total impedance, given in Eq. (SR2) [Eq. (9) in the second-round revised manuscript].

$$I \left(R + \frac{1}{j\omega C} + j\omega L \right) = U_{ind} \quad (SR2)$$

where R , C , and L , are the circuit parameters of the NEMR structure. $U_{ind}(\omega t)$ is the induced voltage of the NEMR structure caused by the flux variation $d\phi$, which can also be defined as Eq. (SR3) [Eq. (10) in the manuscript]. The flux variation $d\phi$ is determined by the flux generated by the transmitter, receiver and/or another NEMR structure.

$$U_{ind}(t) = N \frac{d\phi}{dt} = \frac{dB}{dt} \sum_{k=1}^N s_k \quad (SR3)$$

where N is the number of turns and s_k is the corresponding equivalent area of each turn of the NEMR structure.

Changes Made: Equations (9) and (10) in the second round revised manuscript and corresponding description have been revised.

Comment-3. Response 16 to my comment 16: a better clarification needs to be performed to pass from (R9) to (R10)

Response-3: Thank you for your suggestion, the description of Eq. (R8) to Eq. (R10) in the first round revision is reorganized as follows and supplemented clarification is highlighted. Eq. (R8) to Eq. (R10) in the first round revision is renumbered as Eq. (SR4) to Eq. (SR7).

The voltage proportion between the primary coil and the secondary coil is determined by the self-inductance, mutual inductance, and current in coils, given in Eq. (SR4).

$$\begin{cases} U_s = j\omega L_p i_p + j\omega M i_s + i_p R_p \\ U_o = j\omega M i_p + j\omega L_s i_s + i_s R_s \end{cases} \quad (SR4)$$

where $U_s, U_o, L_p, L_s, i_p, i_s, R_p, R_s$ are the voltage, self-inductance, current, and resistance of the primary side and secondary side, respectively; M is the mutual inductance between two coils.

In the no-load operating condition, the secondary current i_s is considered as zero. The relationship between the input voltage U_s and output voltage U_o can be found as follows.

$$\frac{U_o}{U_s} = \frac{j\omega M i_p + j\omega L_s i_s + i_s R_s}{j\omega L_p i_p + j\omega M i_s + i_p R_p} = \frac{j\omega M}{j\omega L_p + R_p} \quad (SR5)$$

In this transformer, as the primary coil and secondary coil share the same magnetic core and magnetic flux path, the magnetic reluctance of the primary inductance and secondary inductance are considered to be almost the same. Besides, the number of turns of the primary coil and secondary coil are the same. Hence, based on the definition of the mutual inductance given in Eq. (SR6), the mutual inductance is considered to be k times the inductance of the primary coil.

$$M = k \sqrt{L_p L_s} \approx k L_p \quad (SR6)$$

The coupling coefficient k is determined by the magnetic core and its value is between 0 to 1. The total magnetic reluctance of the NEMR structure-based transformer is defined as R_{mt} , consisting of the magnetic reluctance of the proposed structure R_{NEMR} and that of air R_{Air} . Based on the definition of inductance $L_p = N^2 / R_{mt}$, and Eq. (SR6), Eq. (SR5) can be rewritten as.

$$\frac{U_o}{U_s} \approx k \frac{j\omega N^2}{j\omega N^2 + R_p R_{mt}} \quad (SR7)$$

Based on Eq.(SR7), as the primary resistance R_p is greater than zero and the coupling coefficient k is less than 1, only if the magnetic reluctance R_{mt} is negative, the proportion between the output voltage U_o and primary voltage U_s could be larger than 1. Hence, if the voltage proportion is greater than 1, the total magnetic reluctance R_{mt} is negative, correspondingly, the proposed structure indicates a negative equivalent magnetic reluctance.

Changes Made: The descriptions of Eq. (9) and Eq. (10) [Eqs. (R9) and (R10) of the first revision] of the manuscript have been revised

REVIEWERS' COMMENTS

Reviewer #2 (Remarks to the Author):

All my concerns have been addressed from the authors. I thank them for their effort.